# Variational assimilation of IASI SO₂ plume height and total-column retrievals in the 2010 eruption of Eyjafjallajökull using the SILAM v5.3 chemistry transport model

Julius Vira[1], Elisa Carboni[2], Roy G. Grainger[2], Mikhail Sofiev[1]

[1] Finnish Meteorological Institute, Erik Palménin aukio 1, FI-00560 Helsinki, Finland
[2] COMET, Atmospheric, Oceanic and Planetary Physics, University of Oxford, Parks Road, Oxford, OX1 3PU, U.K.

Correspondence to: J. Vira, julius.vira@fmi.fi

## Abstract

This study focuses on two new aspects on inverse modelling of volcanic emissions. First, we derive an observation operator for satellite retrievals of plume height, and second, we solve the inverse problem using an algorithm based on the 4D-Var data assimilation method. The approach is first tested in a twin experiment with simulated observations and further evaluated by assimilating IASI SO₂ plume height and total column retrievals in a source term inversion for the 2010 eruption of Eyjafjallajökull. The inversion resulted in temporal and vertical reconstruction of the SO₂ emissions during 1-20 May, 2010 with formal vertical and temporal resolutions of 500 m and 12 hours.

The plume height observation operator is based on simultaneous assimilation of the plume height and total column retrievals. The plume height is taken to represent the vertical centre of mass, which is transformed into the first moment of mass (centre of mass times total mass). This makes the observation operator linear and simple to implement. The necessary modifications to the observation error covariance matrix are derived.

Regularisation by truncated iteration is investigated as a simple and efficient regularisation method for the 4D-Var based inversion. In the twin experiments, the truncated iteration was found to perform similarly to the commonly used Tikhonov regularisation, which in turn is equivalent to a Gaussian a priori source term. However, the truncated iteration allows the level of regularisation to be determined a posteriori without repeating the inversion.

In the twin experiments, assimilating the plume height retrievals resulted in a 5-20% improvement in root mean squared error but simultaneously introduced a 10-20% low bias on the total emission depending on assumed emission profile. The results are consistent with those obtained with real data. For Eyjafjallajökull, comparisons with observations showed that assimilating the plume height retrievals reduced the overestimation of injection height during individual periods of 1-3 days, but for most of the simulated 20 days, the injection height was constrained by meteorological conditions, and assimilation of the plume height retrievals had only small impact. The a posteriori source term for Eyjafjallajökull consisted of 0.29 Tg (with total column and plume height retrievals) or 0.33 Tg (with total column retrievals only) erupted SO₂ of which 95% was injected below 11 or 12 km, respectively.

## 1    Introduction

Sulphur dioxide ($SO_2$) is one of the major gas-phase species released in volcanic eruptions. Large $SO_2$ releases pose a hazard to aviation, decrease air quality, and as precursors to sulphate aerosols, have a potential impact on Earth's radiative balance (Bernard and Rose, 1990; Robock, 2000; Schmidt et al., 2015). Volcanic $SO_2$ plumes can be detected by satellite instruments measuring in either ultraviolet (UV) or infrared (IR) wavelengths - however, reliably forecasting the atmospheric transport of volcanic plumes is hindered by the lack of in-situ measurements to characterise the emission fluxes of volcanic species (Carn et al., 2009; Stohl et al., 2011; Zehner, 2012).

While methods based purely on satellite retrievals (Theys et al., 2013 and references therein) exist for inferring the total $SO_2$ flux for a given eruption, a successful prediction of volcanic tracers generally requires information on the vertical profile of emissions. An important technique for assessing both vertical and temporal distribution of the emission fluxes is provided by inverse dispersion modelling, first demonstrated for volcanic emissions by Eckhardt et al. (2008).

Inverse modelling of volcanic emissions has been based on using total column retrievals of $SO_2$ or volcanic ash together with a Lagrangian (Kristiansen et al., 2010; Stohl et al., 2011) or Eulerian (Boichu et al., 2013; Boichu and Clarisse, 2014) dispersion models. In addition, Flemming and Inness (2013) devised a trajectory based scheme to evaluate the vertical emission profile, which was used together with assimilation of $SO_2$ retrievals with the IFS (Integrated Forecast System) weather prediction system.  The previous studies have demonstrated that the vertical distribution of emissions can be inferred from total column retrievals in presence of sufficient vertical wind shear. However, in the case of the Eyjafjallajökull eruption in 2010, both Boichu et al. (2013) and Flemming and Inness (2013) pointed out a lack of wind shear and a subsequent difficulty at estimating the vertical distribution of emissions.

Retrievals of $SO_2$ plume height have been performed with various satellite instruments (Carboni et al., 2012; Rix et al., 2012). Nevertheless, only a few studies have incorporated these data into models. Wang et al. (2013) derived a three-dimensional $SO_2$ distribution from retrievals by the Ozone Monitoring Instrument (OMI), and used the distribution to initialize CTM simulations for the 2008 eruption of Kasatochi. Wilkins et al. (2015) used 1D-Var ash retrievals for initialising dispersion simulations. However, neither of the studies used plume height retrievals in inverse modelling of volcanic emissions.

The first objective of the present paper is to assess the usefulness of assimilating $SO_2$ plume height retrievals from the Infrared Atmospheric Sounding Interferometer (IASI) instrument in a source term inversion. Throughout this paper, the term plume height will refer to the vertical centre of mass, which is consistent with the IASI retrievals of this study. Following this definition of plume height, we introduce in Section 3.2 an observation operator for the vertical centre of mass.

Since the observation operator only depends on the centre of mass and column loading, the vertical profile is only partly constrained. However, in contrast to the previous studies, this approach makes no further assumptions about the shape or thickness of the $SO_2$ layer. This could be advantageous, since volcanic ash or $SO_2$ layers vary considerably in depth (Dacre et al., 2014) and can be emitted in multiple, overlapping layers (Kristiansen et al., 2010). Although the variability of the

vertical profiles may introduce uncertainty into the retrieval of the plume height, by assimilating only the centre of mass, we avoid forcing the model into a prescribed vertical profile whose uncertainty may be difficult to quantify. In contrast, our approach makes full use of the retrieval error estimates provided with the IASI data for both column mass and plume height, including the estimated correlation between plume height and mass errors.

The second objective of this paper is to explore the connection between the source term inversion and the 4D-Var data assimilation widely used in numerical weather prediction. Elbern et al. (2000) showed that the 4D-Var assimilation method (Le Dimet and Talagrand, 1986) can be easily extended into estimating emission fluxes with a chemistry transport model. Elbern et al. (2007) further evaluated the joint estimation of emission flux and airborne concentration as a strategy for improving air quality forecasts. However, in this study, the 4D-Var method is formulated to include only the emission forcing, which results in a least squares problem similar to that solved by many existing inversion algorithms. The iterative solution employed in 4D-Var favours a different regularisation approach, which is in Section 4 compared to a more classical regularisation technique.

Finally, we test the variational inversion method and assimilation of plume height retrievals for estimating temporal and vertical distribution of $SO_2$ emission during the 2010 eruption of Eyjafjallajökull. Results of the inversion, presented in Section 5, indicate that although the vertical distribution of emissions is mostly constrained by the total column retrievals and the meteorological conditions, assimilation of plume height retrievals results in more vertically concentrated emission profile. In particular, emissions above 8-10 km between 5 and 9 May 2010 are reduced substantially which is consistent with the observations of the eruption column height as well as the IASI retrievals.

## 2    Model setup and observational data

### 2.1    Dispersion model

The transport and removal of $SO_2$ was evaluated using the dispersion model SILAM (System for integrated modelling of atmospheric composition; Sofiev et al., 2015, http://silam.fmi.fi) version 5.3. The model includes chemical removal of $SO_2$ as described by Sofiev (2000) with the OH climatology of Spivakovsky et al. (2000). The computations were driven by the ERA-Interim meteorological reanalysis (Dee et al., 2011) except for evaluating the simulated satellite retrievals described in Section 4, where operational ECMWF forecasts were used.

SILAM includes a variational data assimilation module, which was previously used for assimilation of air quality monitoring data of $SO_2$ by Vira and Sofiev (2012). The same 4D-Var implementation, including the adjoint codes, is used in this study, but instead of estimating a refinement for a regional emission inventory, we seek to reconstruct the emissions for a single volcanic eruption as a function of time and injection height.

The model was configured for a domain covering 50°E to 30°W and 30°N to 80°N. Horizontal resolution of 0.5° was used for the inversion, while the a posteriori simulations were run with a higher 0.25° resolution.  The vertical grid consists of 34 terrain-following z-levels with a 500 m resolution at the top of the domain increasing to 50 m near the surface.

## 2.2  The IASI dataset

IASI is an infrared Fourier transform interferometer that measures in the spectral range 645–2760 $cm^{-1}$ with spectral sampling of 0.25 $cm^{-1}$ (apodized spectral resolution of 0.5 cm-1) and has global coverage every 12h. The lev1b dataset from EUMETSAT/CEDA archive is used in this study.

The algorithm and the IASI $SO_2$ dataset (column amount and altitude) are explained in more detail by Carboni et al. (2012). The same algorithm has been applied to other volcanic eruptions and successfully compared with other datasets (Carboni et al., 2016; Fromm et al., 2014; Koukouli et al., 2014; Schmidt et al., 2015; Spinetti et al., 2014).

The main points of the retrieval scheme are:

Retrievals are performed for the pixels that were identified by the $SO_2$ detection scheme (Walker et al 2011, 2012).

All the channels between 1100-1200 and 1300-1410 $cm^{-1}$ are used in the iterative optimal estimation retrieval scheme to obtain $SO_2$ column amount and altitude of the plume (in pressure, under the assumption that the vertical concentration of $SO_2$ follows a Gaussian distribution) together with the surface temperature. The scheme determines the column amount and altitude (mean of a Gaussian profile) of the $SO_2$ plume with high precision (up to 0.3 DU error in $SO_2$ amount if the plume is near the tropopause), and it is well suited for plumes in lower troposphere.

The IASI $SO_2$ retrieval is not affected by underlying cloud. If the $SO_2$ is within or below an ash or cloud layer its signal will be masked and the retrieval will underestimate the $SO_2$ amount. In the case of ash this is discernible a posteriori by the value of the cost function. The altitude retrieved for the Eyjafjallajökull eruption plume (using the same dataset as in this paper) in the presence of underlying cloud is consistent with the CALIPSO vertical backscatter profile (Carboni et al 2016, Figs. 1,2,3).

A comprehensive error budget for every pixel is included in the retrieval. This is derived from an error covariance matrix $S_\varepsilon$ that is based on the $SO_2$-free climatology of the differences between the IASI and forward modelled spectra.

Note that the error covariance, $S_\varepsilon$, is defined to represent the effects of atmospheric variability not represented in the forward model, as well as instrument noise. This includes the effects of cloud and trace-gases which are not explicitly modelled. The matrix is constructed from differences between forward model calculations (for clear-sky) and actual IASI observations for wide range of conditions, when we are confident that negligible amounts of $SO_2$ are present. It follows that a rigorous error propagation, including the incorporation of forward model and forward model parameter error, is built into the system, providing quality control and error estimates on the retrieved state. The retrieval state error covariance matrix, used for the assimilation in this work, is directly provided as output of the retrieval pixel by pixel.

## 2.3  Other observations

Section 5 presents comparisons of the a posteriori simulation and the source term with the IASI plume height and total column observations. However, additional datasets required used for evaluating vertical structure of the inversion results. Due to the scarcity of vertically resolved $SO_2$ data, the comparison is based on aerosol observations. The vertical profiles of

the emitted plumes are compared with the backscatter profiles by a satellite-borne lidar, and the $SO_2$ injection height is compared to plume top time series obtained with a C-band weather radar. The potentially different emission and transport of volcanic ash and $SO_2$ introduces some ambiguity to the comparisons; however, as found in Section 5, the different data sources together with the IASI retrievals nevertheless form a fairly coherent picture. This supports the conclusion of Thomas and Prata (2011), who found that ash and $SO_2$ were mostly collocated with each other during the Eyjafjallajökull eruption.

The Cloud-Aerosol Lidar with Orthogonal Polarization (CALIOP) instrument (Winker et al., 2009) on board the CALIPSO satellite is near-nadir viewing, two-wavelength, polarisation-sensitive lidar. The comparisons in this study are shown for the 532 nm total backscatter. Hence, two main challenges are involved in using lidar data for evaluation of simulated $SO_2$ plumes. First, the comparison relies on the assumption that the $SO_2$ plume is collocated with an aerosol plume consisting either of primary particles (mainly volcanic ash) emitted in the eruption, or secondary particles (mainly sulphates) formed during the transport. Second, the volcanic plumes need to be distinguished from water or ice clouds. Although the vertical feature mask available with the CALIOP products provides a classification of aerosol and cloud types, as pointed out Liu et al. (2009) and Winker et al. (2012), thick volcanic ash plumes are frequently misclassified as ice clouds by the standard algorithm.

The comparisons shown in Section 5 and Appendix A consist of CALIOP overpasses intersecting the simulated Eyjafjallajökull plumes. Cases where the CALIOP track is parallel to the plume are omitted, because this makes the profiles extracted from the model very sensitive to horizontal displacement errors. Three of the CALIOP profiles have been collocated with the IASI retrievals under the criteria of less than 2 h time difference and less than 150 km horizontal displacement. The three collocated CALIOP tracks were previously analysed for $SO_2$ by Carboni et al. (2016) along with two additional ones for May 14 and 16; these tracks only intersected the edge of the $SO_2$ plume and did not offer a useful comparison with the model.

The estimated $SO_2$ injection height is compared to the observations of plume top described by Arason et al. (2011). The dataset includes two plume top time series, one estimated from a C-band weather radar located at the Keflavik airport 155 km from the volcano, and one estimated from imagery taken with a web camera located 34 km from the volcano. The 5-minute radar data and the hourly web camera data are averaged in time to facilitate the comparison with the estimated emission. The radar data include values which indicate presence of a plume below the lowest observed height, and in order to maintain consistency with the published 6-hourly time series (Arason et al., 2011; Petersen et al., 2012a), and to avoid a high bias in the averaged values, the altitude of 2.5 km above sea level is assigned to these points.

Both datasets represent the highest altitude with measurable signal from the volcanic plume, and thus, the observed plume height might differ from the midpoint of the emitted layer. The radar data are consequently compared with 80th and 95th percentiles (altitudes with 80 or 95 % mass emitted below) of the emission.

**2.4 Inversion experiments**
The inversion algorithm is evaluated with two sets of experiments based on the eruption of Eyjafjallajökull in 2010,
described in detail by Gudmundsson et al. (2012). The experiments covered the time between 1 May and 21 May, 2010,
which as shown by Flemming and Inness (2013) included the most significant $SO_2$ releases.
The observation operator and the variational inversion technique were first evaluated in experiments with synthetic data
(Section 4), where the simulated observations mimicking the IASI retrievals are extracted from a model simulation. The
simulations are repeated for several assumed artificial source terms. The synthetic experiments evaluate the impact of
assimilating plume height retrievals in addition to total columns, and additionally compare two options for regularising the
inverse problem.
The IASI data were subsequently assimilated to invert for the $SO_2$ emissions in the Eyjafjallajökull eruption. The
inversion was performed both with and without assimilation of the plume height retrievals keeping the setups otherwise
identical.
In all inversion experiments, the emission flux density (kg m$^{-1}$ s$^{-1}$) was estimated for each model level in steps of 12
hours. The model setup used in the synthetic experiments was otherwise identical to that used with the IASI data, but a lower
vertical resolution of 1 km was used to reduce the computational cost.
**3     Assimilation and inversion methods**
The forward problem for volcanic tracer transport is defined by the advection-diffusion equation: given the emission
forcing $f$ , solve

177 (1)
$$\frac{\partial c}{\partial t} + \nabla \cdot (c\mathbf{V}) - \nabla \cdot (K\nabla c) = f(x,t) - s(c,x,t),$$

where $c$ is the tracer concentration, $\mathbf{V}$ is the wind vector, $K$ is the turbulent diffusivity tensor, and $s(c,x,t)$ denotes the
chemical and other sinks, which in this study include the wet and dry deposition of $SO_2$ and its chemical conversion to $SO_4$.
**3.1     Variational source term inversion**
The inverse problem discussed in this paper is to determine the forcing $f$ , given a set of observations depending on $c$ . We
assume that Eq. (1) has been discretised, and following the common notation in data assimilation literature, we denote the
tracer concentrations, collectively for all time steps, with the state vector $\mathbf{x}$ . The state vector is related to the unknown
parameter vector $\mathbf{f}$ by the model operator $\mathcal{M}$ , and to the observations $\mathbf{y}$ by the observation operator $\mathcal{H}$ as $\mathbf{y} = \mathcal{H}(\mathbf{x}_t) + \epsilon$ ,
where $\mathbf{x}_t$ denotes the true state. The random vector $\epsilon$ includes the effect of observation errors as well as the possible
representativeness or model errors associated with $\mathcal{H}$ .
If the errors follow a multivariate normal distribution with covariance matrix $\mathbf{R}$, then a solution to the inverse dispersion
problem can be sought by maximising the likelihood function, which is equivalent to minimising the cost function

189  (2)
$$J(\mathbf{f}) = \frac{1}{2}(\mathbf{y} - \mathcal{H}(\mathbf{x}))^T \mathbf{R}^{-1}(\mathbf{y} - \mathcal{H}(\mathbf{x})),$$

where $\mathbf{x} = \mathcal{M}(\mathbf{f})$.
The cost function assumes that the airborne concentrations, which comprise the state vector $\mathbf{x}$, are completely determined
by the emission. Therefore, contrary to chemical data assimilation studies such as Elbern et al. (2007), no term
corresponding to the concentration in the beginning of assimilation is included. This is reasonable, since the inversion is
performed in a single step, and the state and observation vectors in Eq. (2) cover the whole simulated period. The total $SO_2$
loading was low in the beginning of the assimilation due to the inactive phase of eruption and initial state was therefore
unlikely to affect the inversion for the emission forcing.
Model errors are not explicitly included in the cost function, as the relation between concentrations $\mathbf{x}$ and the emission $\mathbf{f}$
is taken as a strong constraint. Arranging the inversion into a sequence of shorter assimilation windows with a background
term for the initial state would relax this constraint at the boundaries of assimilation windows. However, this would still not
allow for model errors arising within the assimilation window, and problematically, the emitted mass would no longer be
conserved between the assimilation windows. Consequently, we use a single assimilation window and adopt the approach of
previous studies (Seibert et al., 2011; Stohl et al., 2011), where the model uncertainty is incorporated to the observation error
covariance matrix $\mathbf{R}$. The form of $\mathbf{R}$ is explained in more detail in Sections 3.2 and 3.3.
If the model and observation operators are linear, represented by matrices $\mathbf{M}$ and $\mathbf{H}$, then (2) becomes a linear least-
squares problem. For volcanic eruptions with a known location, the emission vector $\mathbf{f}$ is zero almost everywhere, which
makes it feasible to evaluate the matrix $\mathbf{HM}$ and solve (2) algebraically. This is the basis for inversion methods of Boichu et
al. (2013), Eckhardt et al. (2008) and Lu et al. (2016).
As an alternative to the algebraic solution, the minimisation problem (2) can be solved with gradient-based, iterative
algorithms, which avoids evaluating the matrix $\mathbf{HM}$. In this study, the cost function is minimized using the L-BFGS-B (the
limited memory Broyden-Fletcher-Goldfarb-Shanno algorithm with bound constraints) algorithm of Byrd et al. (1995) which
allows constraining the solution to non-negative values. Evaluating the gradient requires solving the adjoint problem for Eq.
(1). The iteration is continued until a stopping criterion is satisfied, e.g. until the norm of the gradient is reduced by a
prescribed factor. However, in Section 4 we will discuss truncating the iteration before formal convergence in order to
control the regularization.
**3.2    Assimilation of plume height retrievals**
Given the tracer concentration $c(x, y, z)$ in three dimensions, the observation operator for column integrated mass $m_{ij}$ is
given by

218    (3)
$$m_{ij} = \sum_{k=1}^{N} w_k c(x_i, y_j, z_k)$$

where $x_i, y_j$ and $z_k$ are the gridpoint coordinates and $w_k$ denotes the thickness (in meters) of the $k$th model level. The layer
concentrations are often weighted with an averaging kernel (Eskes and Boersma, 2003) to account for the vertically varying
sensitivity of the satellite retrieval. In this work, weighting is not applied because the IASI retrievals treat the plume height
explicitly.
In the retrievals, the plume height is represented by its centre of mass

224    (4)
$$Z_{CM,ij} = \frac{1}{m_{ij}} \sum_{k=1}^{N} z_k w_k c_{ijk}.$$

It would be possible to develop an observation operator for $Z_{CM}$, however, the operator would be nonlinear and only defined
for nonzero columns. These problems can be overcome by replacing the centre of mass with the first moment of mass $mZ_{CM}$.
Then, the observations consist of pairs $(m_{ij}, m_{ij} Z_{CM,ij})$ given by

228    (5)
$$\begin{pmatrix} m_{ij} \\ m_{ij} Z_{CM,ij} \end{pmatrix} = \begin{pmatrix} \sum_{k=1}^{N} w_k c_{ijk} \\ \sum_{k=1}^{N} z_k w_k c_{ijk} \end{pmatrix},$$

where $z_k$ is the height of the $k$th model level and $i$ and $j$ refer to the horizontal coordinates. Transforming the observations of
$Z_{CM}$ into $mZ_{CM}$ changes the magnitudes of observation errors, and introduces a correlation between the observation
components $m$ and $mZ_{CM}$. However, this effect can be evaluated and included into the observation operator.
The mean and standard deviation of $m$ and $Z_{CM}$ are denoted as $\mu_m, \sigma_m$ and $\mu_z, \sigma_z$ respectively. Assuming that the
errors of $m$ and $Z_{CM}$ are normally distributed, it can be shown that the variance of first moment equals

234    (6)
$$\begin{aligned} \mathrm{Var}[mZ_{CM}] = {}& \mu_m^2 \sigma_z^2 + \mu_z^2 \sigma_m^2 + \sigma_m^2 \sigma_z^2 \\ & + 2\mu_m \mu_z \mathrm{Cov}[m, Z_{CM}] \\ & + \mathrm{Cov}[m, Z_{CM}]^2. \end{aligned}$$

Under similar assumptions, the covariance of $m$ and $mZ_{CM}$ becomes

236    (7)
$$\mathrm{Cov}[m, mZ_{CM}] = \sigma_m^2 \mu_z + \mu_m \mathrm{Cov}[m, Z_{CM}].$$

Finally, the expectation of $mZ_{CM}$ is shifted due to the correlation between retrievals of $m$ and $Z_{CM}$:

238    (8)
$$\mathrm{E}[mZ_{CM}] = \mu_m \mu_Z + \mathrm{Cov}[m, Z_{CM}]$$

The retrieval errors of different pixels are assumed to be uncorrelated. The observation error covariance matrix **R** is
therefore block-diagonal, and its entries can be evaluated using Eqs. (6) and (7) from the retrieval error estimates $\sigma_m, \sigma_z$
and $\text{Cov}[m, Z_{CM}]$, which are all included in dataset used in this study. However, even if the standard deviations are known
accurately, the means $\mu_m$ and $\mu_z$ need to be substituted with the observed values of $m$ and $Z_{CM}$. The impact of this
approximation is evaluated numerically in Section 4.
Assimilation schemes commonly assume uncorrelated and unbiased observation errors. A non-diagonal $\mathbf{R}$ can be
introduced with a transformation of variables: define

246     (9)
$$\mathbf{L}^T\mathbf{L} = \mathbf{R}^{-1}$$
$$\tilde{\mathbf{y}} = \mathbf{L}(\mathbf{y} - \mathbf{b})$$
$$\tilde{\mathbf{H}} = \mathbf{L}\mathbf{H}$$

where $\mathbf{L}^T\mathbf{L}$ is the Cholesky factorisation of the inverse observation error covariance matrix $\mathbf{R}^{-1}$ and $\mathbf{b} = \left(0, \text{Cov}[m, Z_{CM}]\right)$
corrects for the bias according to Eq. (8). Then, substituting the transformations of Eq. (9) into the cost function (2) shows
that assimilation of $\mathbf{y}$ with the original $\mathbf{R}$ is equivalent to assimilation of $\tilde{\mathbf{y}}$ using the transformed observation operator $\tilde{\mathbf{H}}$
with unit matrix in place of $\mathbf{R}$.
The above formulas can be implemented as a preprocessing step for the observations. In summary, the procedure is then
as follows:
1.  For each available IASI pixel $i$, evaluate the tuple $\mathbf{y}_i - \mathbf{b}_i = (m_i, m_i Z_{CM,i} - \text{Cov}[m_i, Z_{CM,i}])$ and the corresponding

254         2x2 covariance matrix $\mathbf{R}_i$.

2.  Factorise $\mathbf{R}_i^{-1} = \mathbf{L}_i^T\mathbf{L}_i$ and transform the observations according to Eq. (9).
3.  Store the transformed observations $\tilde{\mathbf{y}}_i$ with their pixel-specific vertical weighting functions given by rows of the

257         matrix $\tilde{\mathbf{H}} = \mathbf{L}_i\mathbf{H}$.

After the transformation, the observations are handled identically to regular column observations with a vertical weighting
function.

## 3.3    Observation errors

The IASI retrievals used in this study include pixel-specific error estimates for total column and plume height retrievals.
The estimates are derived statistically (Carboni et al., 2012) from differences between the transmission spectra computed by
a forward model and those observed by IASI. Together with estimates for the correlation between plume height and total
column retrieval errors, this provides the necessary input for equations (6) and (7).
The retrieval error estimates are only provided for pixels with positive $SO_2$ detection. For the non-$SO_2$ pixels, which are
assimilated as zero values, a different estimate is used, based on the detection limits estimated by Walker et al. (2012). The
detection limit was translated into a standard deviation of a Gaussian random variable assuming, conservatively, a
probability of 0.95 for a correct detection.
However, performing the inversions with **R** defined only by retrieval errors resulted in poor a posteriori agreement with
the IASI data, which suggested that the retrieval errors are not sufficient to describe the discrepancy between the simulated
and observed values. As will be shown with the synthetic experiments, the impact of model uncertainty is significant
compared to the retrieval errors, and it needs to be taken into account. The problem of model errors affecting the inversion is
discussed by Boichu et al. (2013), who found the impact to depend strongly on treatment of zero-value observations, and
consequently chose to keep only every tenth zero-valued observation.
In this study, the model errors are included by modifying the observation error covariance matrix, which is set to
$\mathbf{R} = \mathbf{R}_{obs} + \mathbf{R}_{model}$, where $\mathbf{R}_{model}$ is constant, diagonal and determined experimentally. The model error standard deviation for
total column observations is set to 2 DU for both the experiments using synthetic data (Section 4) and for the inversion for
Eyjafjallajökull (Section 5), while the model error for the plume height retrievals was set to 2 km for the synthetic
experiments and 1 km for the Eyjafjallajökull inversion. Reducing the plume height standard deviation to 1 km in the
synthetic experiments resulted in large negative bias in the total emission, while increasing the standard deviation to 2 km
did not significantly change the total emission in the inversion for Eyjafjallajökull.
The model errors for plume height and total column are assumed uncorrelated and independent of the observation errors.
However, their effect is propagated to the covariance matrix for first moment according to Eq. (6) . The actual model errors
evolve dynamically and are likely to be variable and correlated in space and between the plume height and total column
components; however, including these effects appears difficult in the current inversion approach.
**3.4 Regularization**
The least squares problem (2) has a unique solution only if the matrix **HM** is of full (numerical) rank. Furthermore, if
**HM** is close to singular, the problem remains ill-posed, which results in a noisy solution. Consequently, some form of
regularisation has been employed in all previous works based on the least-squares approach.
A common option is the Tikhonov regularisation (Tikhonov, 1963; Engl et al., 2000), which introduces a penalty term
into the cost function (2), which in the simplest form becomes

292 (10)
$$J(\mathbf{f}) = \frac{1}{2}(\mathbf{y} - \mathbf{Hx})^T \mathbf{R}^{-1}(\mathbf{y} - \mathbf{Hx}) + \alpha^2 \sum_{k,n} w_k \mid f_{k,n} \mid^2$$

where the summation is over levels $k$ and timesteps $n$. The weights $w_k$ in Eq. (10) are set equal to the thickness of each
model layer; this makes the penalty term consistent with its continuous counterpart $\int f(z,t)^2 \, dt dz$, which in turn ensures that
the regularisation term does not depend on the vertical discretisation.
The penalty term can be modified to include a non-zero a priori source term. However, this approach is not taken in the
present work. Instead, we aim to choose the level of regularisation optimally, so as to avoid excessive bias in the regularised
solution. The need for regularisation depends on the coverage of observations, accuracy of the forward model as well as on
the meteorological conditions controlling the dispersion. Thus, the regularisation parameter $\alpha^2$ cannot be chosen a priori.
In this work, a criterion known as the L-curve (Hansen, 1992) is used for determining the amount of regularisation. In the
L-curve approach, the inversion is performed with various values of $\alpha^2$, and the residual $\|y - Hx\|$ is plotted as a function of
the solution norm $\|f\|$. For ill-posed inverse problems, the curve is typically L-shaped. The residual initially reduces quickly
as the regularization is relaxed, however, for some value of $\alpha^2$, the curve flattens and reducing the regularization further
only marginally improves the fit. This point, where L-curve reaches its maximum curvature, is taken to represent the optimal
regularisation. In the present study, the L-curve is evaluated without the frequently used logarithmic transformation.
The main advantage of the L-curve method is that it does not rely on a priori estimates for the observation error. This is
useful, since in practice the discrepancy between simulated observations and the data is also affected by model errors, which
are poorly known. The L-curve was, in effect, used in inverse modelling of volcanic $SO_2$ also by Boichu et al. (2013).
Changing the regularisation parameter requires the minimisation to be started over, which is costly in the variational
inversion scheme where each iteration requires a model integration. However, as noted by Fleming (1990) and Santos
(1996), the iteration itself forms a sequence of solutions with decreasing regularisation. Thus, instead of minimising the
regularised cost function (10), we iterate to minimise the original cost function (2), and truncate the iteration according to the
L-curve criterion. This approach, known as regularisation by truncated iteration (Kaipio and Somersalo, 2006), or iterative
regularisation (Hansen, 2010), provides a computationally efficient method to regularise large-scale inverse problems. In the
following section, we show experimentally that the truncated iteration results in similar solutions for the source term
inversion as the more common Tikhonov regularisation.

## 4   Experiments with synthetic data

Regularisation by truncated iteration has been studied in detail especially for Krylov subspace based algorithms (Calvetti
et al., 2002; Fleming, 1990; Kilmer and O'Leary, 2001). The effect of truncated iteration on quasi-Newton minimisation
methods, such as the L-BFGS-B algorithm used in this work, has been studied less extensively. To evaluate the truncated
iteration in comparison to Tikhonov regularisation for inverse modelling of volcanic emissions, we performed an experiment
with synthetic observations extracted from forward model simulations. In addition to the comparison of regularisation
methods, the synthetic experiments enable us to evaluate robustness of the L-curve method and to assess the impact of
assimilation of plume height retrievals, and to quantify how model errors affect the source term estimate.
For the sake of computational convenience, the experiments in this section are not performed using the variational method
described in Section 3.1, but instead the forward sensitivity matrix **HM** is evaluated by running a separate model simulation
for each component of the emission vector **f**. The sensitivity matrix is subsequently used for evaluating the cost functions
(Eq. (2) for truncated iteration, Eq. (10) for Tikhonov regularisation) and the respective gradients required by the L-BFGS-B
minimisation code. Evaluating the sensitivity matrix also provided an opportunity to numerically confirm the equivalence of
the matrix-based and variational inversion methods.
The experiments with synthetic data were set up for the same time (1 to 20 May, 2010) as the inversion for
Eyjafjallajökull. The synthetic observations were evaluated by running forward simulations with a set of artificial source
profiles (cases A to D) shown in the leftmost column of Figure 1. The synthetic observational data (total columns and first
moments as explained in Section 3.2) correspond to the locations and times covered by the IASI overpasses during the
simulated period.
The artificial source terms A and B are defined arbitrarily, while cases C and D are realisations of a stochastic process
where the total flux (kg/s) is given by a lognormal, temporally correlated random variable and the eruption height follows the
relation of Mastin et al. (2009). At each time, a piecewise constant vertical profile is assumed with a transition at 75% of
height. The emission rate is distributed evenly between the two sections.
The simulations with artificial source terms were driven by the meteorological data valid for the simulated period. Two
sets of meteorological input were used: the synthetic observations were generated using the operational ECMWF forecast
fields, but to simulate the effect of model errors, the sensitivity matrix used in the inversions was evaluated using the ERA-
Interim as the meteorological driver. Although changing the meteorological driver does not cover all sources of model error,
we expect the resulting perturbation to have statistical properties similar to the real model uncertainty.
The effect of retrieval errors was simulated by perturbing the extracted (simulated) observations with additive Gaussian
noise. In order to perturb the simulated plume height retrievals, the unperturbed simulated first moments and total columns
were first converted back to the centre of mass and total column for the pixels with column density higher than 0.2 DU in the
forward run. Then, both the simulated centre of mass and the total column were perturbed and transformed back to the
(perturbed) total columns and first moments. The total columns were perturbed with standard deviation equal to 0.1 DU + 10
% of the true value; the centres of mass were perturbed with a constant standard deviation of 1 km. A negative correlation
coefficient of -0.9 was assumed between the perturbations to the total column and centre of mass.
The error covariance matrix used in the inversion was supplemented with 2 DU and 2 km "model error" as described in
Section 3.3. For the inversions using simulated plume height retrievals, the observation error covariance matrices were
transformed according to Eqs.(6) –(8) using the perturbed centre of mass and total column values for $\mu_z$ and $\mu_m$ .
The residual and solution norms, which define the L-curves, are evaluated consistently to the penalized cost function (10):

356    (11)
$$\|\mathbf{Hx}-\mathbf{y}\| = \sqrt{(\mathbf{Hx}-\mathbf{y})^T \mathbf{R}^{-1}(\mathbf{Hx}-\mathbf{y})}$$
$$\|\mathbf{f}\| = \sqrt{\sum_{k,n} w_k \mid f_{k,n} \mid^2}$$

where $\mathbf{f}$ denotes the emission, $\mathbf{x} = \mathbf{Mf}$ and $w_k$ is the thickness of the $k$th model layer. To evaluate the L-curve for
Tikhonov-regularisation, the parameter $\alpha^2$ was incremented in discrete steps given by $\alpha_i^2 = 10^7 \cdot 2^{-i}$ for $i = 0,1,2,...$ . The L-
BFGS-B minimization method with non-negativity constraint was used for both Tikhonov regularisation and the truncated
iteration; in the case of Tikhonov regularisation, the iteration was continued for each $\alpha_i^2$ either until convergence or for
maximum of 50 iterations. A zero-valued solution was always used as the first guess in the iteration. With the truncated
iteration, the weights $w_k$, required by Eqs. (10) and (11), are not explicitly included in the cost function. Instead, the same
effect is achieved by transforming the parameter vector as $f'_{k,n} = w_k^{1/2} f_{k,n}$.
The point where the L-curve flattens, which is taken as the final solution, was determined numerically. First, the points
$(\|\mathbf{f}\|, \|\mathbf{Hx} - \mathbf{y}\|)$ are sorted according to increasing $\|\mathbf{f}\|$. Then, the points where the residual increases are removed, and finally,
the optimal point is chosen using the "triangle" algorithm of Castellanos et al. (2002).
Figure 1 presents the inversion results using Tikhonov regularisation with total column observations, truncated iteration
with total column observations, and truncated iteration with total column and plume height observations. Regardless of the
assumed source term or inversion method, the emission timing is well captured within the 12 h resolution. The overall
vertical profiles are also recovered, however, spurious features are present especially in cases B and C.
For comparison, Figure 2 presents the solution corresponding to the case B in Figure 1 but evaluated without model errors
– that is, using the same sensitivity matrix $\mathbf{HM}$ for both evaluating the observations and performing the inversion. In this
case, regularisation was not needed, and the true solution was recovered almost perfectly despite the noisy observations.
Thus, the noise present in the estimated solutions in Figure 1 is mainly due to model error, which affects the elements of
matrix $\mathbf{M}$. All other results presented in this section are obtained in presence of model errors.
Numerical evaluation of the inversion results in terms of RMSE and relative bias is presented in Table 1. The scores are
evaluated for both truncated iteration and Tikhonov regularisation, each with and without plume height observations.
Furthermore, two numbers are given for each case: the optimal value, corresponding to the regularisation (for Tikhonov, the
value of $\alpha^2$, for truncated iteration, the iteration number) with lowest RMSE, and the L-curve value corresponding to the
choice of regularisation as determined from the L-curve explained above. Clearly, the regularisation with optimal RMSE is
not necessarily optimal with respect to bias.
For all cases, the optimally truncated iteration had lower RMSE than the optimally tuned Tikhonov regularisation.
However, this advantage was not always realised when the truncation was determined from the L-curves, which are shown in
Figs. 3 and 4. For the Tikhonov regularisation, the L-curve solution was generally closer to the optimal. The difference is
caused by differing features of the L-curves for the two regularisation methods: for the Tikhonov regularisation, the L-curve
forms a convex graph varying smoothly with $\alpha^2$, while the curves formed by the L-BFGS-B iterates are neither smooth nor
even monotonous. Although points where the residual increases are omitted from the search, points with a locally large
curvature remain in the curve, and such points are responsible for the under-regularised L-curve solutions in cases A and D
when only total column was assimilated.
In Figs. 3 and 4, the root mean squared error (RMSE) of the solution is shown next to each L-curve as a function of the
regularisation parameter. As expected, the RMSE initially drops as the regularisation is relaxed, reaches a minimum, and
eventually increases as the solution becomes contaminated by noise. This behaviour was especially clear when only total
column observations were assimilated. When also centres of mass were assimilated, the minima in RMSE become weaker,
and the RMSE with maximum number of iterations was only slightly higher than optimal. Thus, assimilating the centres of
mass had the unintended but potentially useful side effect of making the inversion less sensitive to under-regularisation.
Since the regularised cost function (10) favours solutions with a small squared norm, the inversion is expected to
underestimate the true emission. If only total column observations are used, the underestimation remains small, being 5 –
10% for the L-curve solutions with truncated iteration, and up to 15 % for the corresponding Tikhonov regularised solutions.
However, when the plume height observations were included, the negative biases increased to 15-25% even when using
truncated iteration.
Magnitude of the negative bias turned out to be sensitive to the assumed model uncertainty as described by the covariance
matrix $\mathbf{R}_{mdl}$ . Reducing the standard deviation for plume height errors to 1 km resulted in negative biases between 25 and
35%. As a further sensitivity test, we evaluated the effect of approximating the true values for total column and plume height
with the respective observed values when transforming the observation error covariance matrix, as explained in Section 3.2.
Using, unrealistically, the true values in the inversion, the relative biases were reduced to 16-21%. The RMSE was reduced
by up to ~15%. It can be noted that none of the tested setups describe an observation error covariance matrix that would
perfectly match the perturbations applied the simulated observations, since the model errors, simulated by using a different
meteorological driver, are not well described by additive, white noise. Taking the cross-correlations and spatial variation of
model errors into account might lead into different optimal $\mathbf{R}_{mdl}$ .
While the experiments in this section were performed by pre-evaluating the matrix $\mathbf{HM}$ , in 4D-Var, the multiplications
by $\mathbf{HM}$ and its transpose are replaced by forward and adjoint model evaluations. Although the approaches are formally
equivalent, this change results in a slightly different sequence of iterations from which the L-curve is evaluated. To
investigate this difference, we performed the inversion using the real IASI data using both approaches. The two solutions are
shown in Figure 5. The total released mass differs by less than 1% between the solutions, and the emission patterns are
qualitatively similar. The differences for individual values, although larger, appear small compared to the inversion errors.
In summary, the experiment with synthetic data showed that the truncated iteration resulted in solutions similar to those
obtained with the more common Tikhonov regularisation. This makes the truncated iteration, in combination with the L-
curve, an attractive option for regularising the variational source term inversion. On the other hand, no regularisation was
needed in absence of model error which indicates that the need for regularisation is likely to also depend on quality of the
forward model. This emphasizes the need for a robust method to determine the appropriate regularisation according to the
situation at hand.
**5    Inversion results for Eyjafjallajökull**
Optimising the source term following the regularisation strategy (truncated iteration) described in Section 3.4 results in
satellite-derived estimates on the temporal and vertical emission profiles, as well as on the total emitted amount. The
solutions presented here correspond to iterates chosen from the L-curve using the algorithm described in Section 3.4. For

assimilation of column mass only, the 9th iterate was chosen; with column mass and plume height assimilation, the 13th iterate was chosen. Similarly to the synthetic experiments, the initial iterate was a zero solution. The L-curves are shown in the supplementary information.

Figure 6 shows the temporal and vertical distribution of the SO$_2$ emission obtained both with and without assimilation of plume height. The plume height time series estimated from radar and camera observations (Petersen et al., 2012b) are plotted on top of the emission distributions. Both the camera and radar observations represent the top of the visible plume, and even if the visible plume does not necessarily coincide with the SO$_2$ plume, the plume height observations provide an indication of the eruption activity.

Figure 7shows the vertical profile of emissions integrated over the whole period. The bulk of emissions are between 2 and 8 km even if only column density is assimilated. Assimilating the plume height retrievals further decreases the fraction of emissions above 8 km. When the plume height is assimilated, about 85% of total emission is estimated below 8 km and about 95% below 11 km. Without assimilation of plume heights, the 95% level raises to 12 km.

The strongest emission occurred during 6th May. However, the vertical distribution of the peak depends on whether the plume height is assimilated. While the maximum occurs at 5-6 km, if plume height is not assimilated, secondary maxima appear at 11 km, reaching 13 km on 9th May. If plume height retrievals are assimilated, the emission above about 8 km is strongly suppressed. Similarly, on 18th May, the isolated emissions at 10 and 15 km are largely removed when the plume height is assimilated.

A more quantitative view on the effect of assimilating the plume height retrievals is given by Figure 8, which compares the estimated centre of mass of the SO$_2$ emission with the retrieved plume heights. The plume heights are shown as averages within both 50 and 500 km radius from the volcano. The averages over wider area have better temporal coverage and they are likely to be less affected by unresolved temporal or spatial variations in the plume height. The retrievals with estimated error larger than 5 km are excluded from the averaging (although used in assimilation).

In addition, Figure 8 includes radar and camera observations of the plume top which are compared with the 80th and 95th percentiles of the emission. The 95th percentile, although formally more representative of the top of emissions, shows very large fluctuations compared to both observations and the 80th percentile, which suggests that the highest percentiles might not be a robust way to characterise the plume top in the inversion results.

Over the whole period, the inversion results show a larger variability of injection height in comparison to both IASI and the radar or camera time series. Between May 4 and 5 and later May 10 and 17, the average IASI retrievals and the emission centre of mass agree mostly within 1-2 km, as do the radar observations with the 80th percentile of emission. An exception is the evening of May 11 when the injection height appears overestimated, however, the total emission rate was low at that time. Assimilation of plume height retrievals had little impact on the injection height during these times.

Between May 6 and 10, the injection height is overestimated in comparison with both IASI and radar observations. Assimilating the plume height retrievals improves the comparison, but the injection height remains 2-5 km too high compared to the averaged IASI retrievals. A similar overestimation occurs on May 17 and 18. Assimilating the plume height

again reduces the overestimation significantly on those days, however, both the centre of mass and the percentiles remain
overestimated.
The total released mass of $SO_2$ is 0.33 Tg when the plume height is not assimilated and 0.29 Tg when the plume height is
assimilated. Figure 8d, which depicts the emission flux as a function of time, shows that while the largest difference in
emission rate is during the peaks of 6th May, the assimilation of plume heights tends to decrease the emission rate
throughout the eruption.
The $SO_2$ column densities simulated a posteriori are shown for 5-7 May in Figure 9 along with the corresponding IASI
retrievals. The overall patterns are well reproduced, although the column density is underestimated for some parts of the
plume, especially on 6th and 7th of May. Due to the smaller total emission, the column densities are slightly lower when
plume height is assimilated. Comparisons of the total columns for all 20 days are presented in the supplementary material.
Figure 10 shows the simulated plume height (evaluated as centre of mass) for 7-9 May, which corresponds to the period
of overestimated injection height shown in Figure 8. Compared to IASI, the inversion using only total columns tends to
overestimate the plume height for all three days. As expected from Figure 8, when the plume height retrievals are
assimilated, the overestimation is reduced, but not entirely removed.
A more detailed evaluation of the vertical profiles is enabled by comparison with the CALIOP lidar backscatter data. It
should be noted that the most prominent features in the CALIOP data are regular clouds; in particular, this includes the near-
constant layers located at 1-2 km altitude.
In Figure 11, the simulated $SO_2$ concentration is plotted as contours together with the CALIOP attenuated backscatter data
collected on May 6 and 8, 2010. On both days, the track segment intersects the $SO_2$ plume near its source. On May 6, this
part of the volcanic plume is obscured by a cloud, but a distinctive aerosol layer is visible south of 60° N. This layer is
reproduced by the model, however, the observed vertical extent is much thinner than modelled, indicating that the vertical
variation of the transport was not sufficient to resolve the emission vertically. The plume height for the thickest part of the
plume is nevertheless reproduced within ~2 km, and hence, assimilating the plume height retrievals had only little impact on
the simulated plume.
On May 8, the highest simulated concentrations coincide with a strong backscatter signal at 3-4 km altitude close to the
emission (near 62° N). The altitude is consistent with the averaged IASI plume height retrievals shown in Figure 8, whereas
the simulated vertical extent between 2 and 7.5 km is again too wide. While a second layer between 8 and 12 km is present
in the CALIOP data, the horizontal extent of this feature is far too wide to represent the volcanic plume. A third simulated
$SO_2$ layer is present at 13 km only if plume height retrievals are not assimilated; this demonstrates the difference of injection
heights seen in Figure 8.
The CALIOP track on May 8 also crosses an older $SO_2$ plume around 48° N, where the simulated vertical extent is
compatible with the CALIOP data. However, a prominent layer extending between 50° and 55° N is present in the CALIOP
data. The layer is classified partly as cloud and partly aerosol in the CALIOP vertical feature mask (not shown), but the layer
does not coincide with the simulated $SO_2$ plume. However, Figures 9 and 10 indicate that the simulated plume was

erroneously displaced towards west during the evening of May 7. Taking this into account, it is feasible that the observed backscatter would be caused by the volcanic plume. The 3-4 km altitude of the layer would agree with the IASI plume height retrievals (Figure 10) and support the below 5 km injection heights indicated by the IASI and radar data in Figure 8.

Figures 12 through 14 combine the simulated $SO_2$ profiles and the CALIOP data with collocated IASI total column and plume height retrievals. The simulated vertical distributions are mostly consistent with both the CALIOP and the IASI data. In Figure 12, the 3-4 km mean altitude of the peak reaching 20 DU according to the IASI data is reproduced by the model. The altitude of the plume extending towards south (between 48-50° N) is also reproduced given the higher retrieval uncertainty. The column densities up to 20 DU, however, are not reproduced: the highest simulated values are displaced towards west and remain below 10 DU.

Figures 13 and 14 show generally similar level of agreement in the vertical structures. In both figures, the northern part of the plume (55-60° N) is partly obscured by a cloud, which is reflected by the large retrieval error estimates. In both figures, assimilating only total column retrievals resulted in several isolated $SO_2$ layers between altitudes of 10-15 km. Presence of these layers is supported by neither IASI nor CALIOP data. Even if the corresponding $SO_2$ emissions did not coincide with ash emissions, some CALIOP signal could be expected due to the sulphate particles forming in the plume. Altogether, the comparisons in Figs. 12 through 14 and the comparison of the emission profiles (Figure 8) support the conclusion that the emissions above 8-10 km on 6-9 May were an artefact and probably related to insufficient wind shear.

Further comparisons with CALIOP data on 14 to 17 May are shown in Appendix A. The simulated vertical distributions generally coincide with layers observed by CALIOP; however, assimilation of plume height retrievals had little impact on the simulated plumes at those times.

## 6   Discussion

No a priori assumptions regarding shape the emission profile were made in this study. The comparison with the IASI retrievals, CALIOP data and weather radar observations of the plume shows that the resulting vertical distributions were frequently in good agreement with the observations even if only total column retrievals were used in the inversion. The most notable exception were the emissions between 6 and 10 May, when the injection height was strongly overestimated, and although assimilating the plume height retrievals improved the agreement, the discrepancy was not fully resolved. Since the plume height retrievals are introduced as a weak constraint, a complete match between the inversion results and the observation data is not expected. However, some of the discrepancies remain too large to be explained by retrieval errors even together with the assumed model 1 km uncertainty.

Generally, two factors could lead to an inaccurate reconstruction of the vertical profile from the total column observations. First, the horizontal transport patterns on different altitudes might be too similar for resolving the vertical structure. Second, the simulated horizontal patterns might be too inaccurate due to errors or low resolution of the transport model or its input data. Since the inversion does not allow for systematic model errors, including the plume height retrievals

in the inversion is expected to improve the vertical profile mainly in the first case. The discrepancy remaining between the observed and modelled plume heights suggests that model errors were at least partly responsible for the overestimation of injection heights on 6-10 May.

The main effect of assimilating the plume height retrievals was the reduction of emissions above 10-12 km. Although these emissions are not large compared to the total emission, this outcome has some qualitative significance, since without assimilation of plume heights, some emissions would be assigned above the tropopause. In addition to the data presented in the previous section, previous studies based on lidar data (Ansmann et al., 2010) or aircraft measurements (Schumann et al., 2011) do not suggest significant injection above the 10 km altitude. However, these studies were mainly focused on volcanic ash instead of $SO_2$. On the other hand, the $SO_2$ plume height estimates derived from the GOME-2 satellite instrument by Rix et al. (2012) do indicate heights above 10 km and up to 13 km on 5th of May. Neither our data nor inverse modelling reproduces this result, as the plume heights retrieved from IASI data are below 6 km for that day, which agrees with the modelled plume heights (not shown) even when only total column retrievals are included in the inversion.

Among the previous emissions estimates for Eyjafjallajökull, Flemming and Inness (2013) estimated a 0.25 Tg total $SO_2$ release using GOME-2 satellite retrievals, and 0.14 Tg using the OMI retrievals. Our estimates of 0.29-0.33 Tg are higher, especially compared OMI, but this is consistent with the higher total $SO_2$ burden estimated (Carboni et al., 2012) from the IASI data used in this study. Using the GOME-2 data, Flemming and Inness (2013) furthermore estimated $SO_2$ injection heights (defined as centres of 2-3 km thick layers) to mostly between 4 and 6 km above sea level with a peak reaching 10 km on May 19th. This agrees reasonably well with our mean profile (Figure 7), although contrary to our results without plume height assimilation, Flemming and Inness (2013) did not obtain the injection heights above 6 km on May 6th and 7th.

Boichu et al. (2013) used the IASI retrievals of Clarisse et al. (2012) to invert for temporally resolved $SO_2$ emissions of Eyjafjallajökull between May 1th and 12th, 2010, and estimated a total emission of about 0.17 Tg. Our inversion yielded for the same time 0.21 (total column and plume height retrievals) or 0.23 (total column only) Tg of $SO_2$. The larger total emission in our study might be due to assumptions regarding plume height in the IASI retrievals. The retrievals used by Boichu et al. (2013) assumed constant 7 km plume height, while the retrieved plume heights in this study were frequently lower especially near the volcano, and this would result in a higher retrieved values for the total column. For the emission, Boichu et al. (2013) assumed a constant injection height of 6 km, which turns out to coincide with the maximum of the mean profile (Figure 7) obtained in this study.

Stohl et al. (2011) determined the temporal and vertical distribution of volcanic ash emissions for the Eyjafjallajökull eruption with an inversion constrained by SEVIRI ash retrievals and an a priori source derived from plume top observations. Although the ash and $SO_2$ emissions cannot be compared quantitatively, the mean vertical profile obtained using ECMWF meteorological data (Fig. 2 in Stohl et al. (2011) is not very different from the one in Figure 7. In both profiles, the emissions are restricted mainly below 8 km and have maxima at 6 km.

Including the plume height retrievals in the inversion resulted in a total emission 12% lower than with total column retrievals only. Similar differences were observed in the experiments with synthetic data discussed in Section 4, where the

inversion results were biased low by 15-20% using both plume height and total column retrievals and by only 2-10% using
total columns only.
In ideal conditions, assimilating the plume height information should not affect the simulated total columns. However,
adding a vertical constraint to the inversion can never improve the agreement for total columns, and in presence of realistic
model uncertainty, a negative effect can be expected. The systematic tendency towards smaller emission may be caused by
the regularisation, which penalises the quadratic norm of the solution. The synthetic experiments indicated that introducing
the plume height retrievals did not allow relaxing the regularisation, since the optimal level (as identified from the parameter
$\alpha^2$ ) was similar with and without the plume height observations.
On the other hand, the synthetic experiments also indicated that the estimation error for the total emission was only
moderately sensitive to the differences of the assumed source terms. The estimate for total emission was also robust with
regard to the vertical resolution, as halving the vertical resolution of the reconstruction (compare Figs. 5 and 6) resulted in
only minimal change in the total emission. The estimated total emission could, nevertheless, be affected by biases in the
satellite retrievals, or by model errors not exposed by the change of meteorological driver.
The experiments with synthetic data furthermore showed that the need for regularisation, or in Bayesian terms, the need
for a priori information, was strongly affected by uncertainty of the forward model. The efforts needed to handle zero-valued
observations in this and other studies (Boichu et al., 2013; Seibert et al., 2011) support this conclusion. The errors arising
from the dispersion model are likely to be correlated in space, and therefore, introducing the corresponding non-diagonal
elements in the error covariance matrix $\mathbf{R}$ could improve the inversion results. While the regularisation used in this work is
equivalent to a zero-valued a priori source, a more informative a priori source could be accommodated with a change of
variable. Other forms of regularisation proposed for the volcanic source term inversion include second-order temporal
smoothing (Boichu et al., 2013), which also could be handled by truncated iteration as discussed by Calvetti et al. (2002).
The variational inversion method is computationally efficient if high temporal or vertical resolution is desired for the
reconstruction. In the current configuration, the reconstructed solution had formally 1360 degrees of freedom. Each iteration
consisting of one forward and one adjoint integration, the 25 iterations would require model integrations equivalent to about
1000 simulated days. In comparison, evaluating the matrix $\mathbf{HM}$ directly would require 1360 model integrations, and if the
sensitivity was evaluated in windows of e.g. 72 hours, almost 4000 simulated days would be required. The matrix-based
approach is, however, more easily parallelised, while the parallelisation of the variational method relies on the dispersion
model. In our configuration, one iteration took about 5 minutes wall clock time on a 20-core node of a Cray XC30
supercomputer.
A drawback of the 4D-Var inversion method is that the a posteriori error covariance matrix for the source term is difficult
to evaluate. However, Monte Carlo techniques could be used to sample the a posteriori uncertainty.

# 7 Conclusions

We have presented an observation operator for retrievals of the vertical centre of mass of a tracer plume. The operator is based on transforming the centre of mass into first moment of mass using the retrieval of total column. The approach was tested by performing a source term inversion using both artificial data and the $SO_2$ retrievals from the IASI instrument during the Eyjafjallajökull eruption in May 2010. The inverse problem was solved with the 4D-Var method embedded into the SILAM dispersion model, and the truncated iteration is proposed as an efficient regularisation method for the 4D-Var inversion. Using both real and synthetic data, the 4D-Var method was shown to produce a similar solution as the more common algebraic method, but at lower computational cost.

The inversion results for Eyjafjallajökull were compared to radar based ash plume observations and CALIOP lidar profiles. The comparisons show that assimilating the plume height retrievals reduced the overestimation of injection height during individual periods of 1-3 days. However, for most of the simulated 20 days, the injection height was constrained by meteorological conditions and assimilation of the plume height retrievals had only small impact.

When the plume height was assimilated, about 85% of the 0.29 Tg total emission was below 8 km and about 95% was below 11 km. Compared to previous modelling studies (Boichu et al., 2013; Flemming and Inness, 2013), the total emission is 15-20% larger taking into account the differences in temporal coverage of the studies.

Introducing the plume height retrievals in the inversion may have an adverse effect on the estimated total emission. In the experiment with artificial observations, the inversions with only total column data had a negative bias of 2-10% which increased to 15-20% when the plume height observations were included. In the inversion for Eyjafjallajökull, performing the inversion using only total column retrievals resulted in ~15% larger total emission, which is consistent with the experiments with simulated observations.

Experiments with both synthetic and real data suggest that the inversion is sensitive to errors in the forward model, and to their assumed uncertainty. Methods more robust to model errors are a topic suitable for future research.

## Acknowledgements

This work has been supported by SMASH and VAST (ESA), EmblA (NordForsk) and EUNADICS-AV (EU H2020). E.C and R.G.G. acknowledge funding from the NERC SHIVA (NE/J023310/1) and VANAHEIM (NE/1015592/1) projects. The work of E.C. has been partly funded by the EC-FP7 APhoRISM project. The authors acknowledge NASA ASDC for provision of the CALIOP data. The authors thank Marje Prank for comments on the manuscript.

## Code availability

The source code for SILAM v5.3, including the data assimilation component, is available on request from the authors (julius.vira@fmi.fi, mikhail.sofiev@fmi.fi).

**Appendix A: Additional comparisons with CALIOP data**

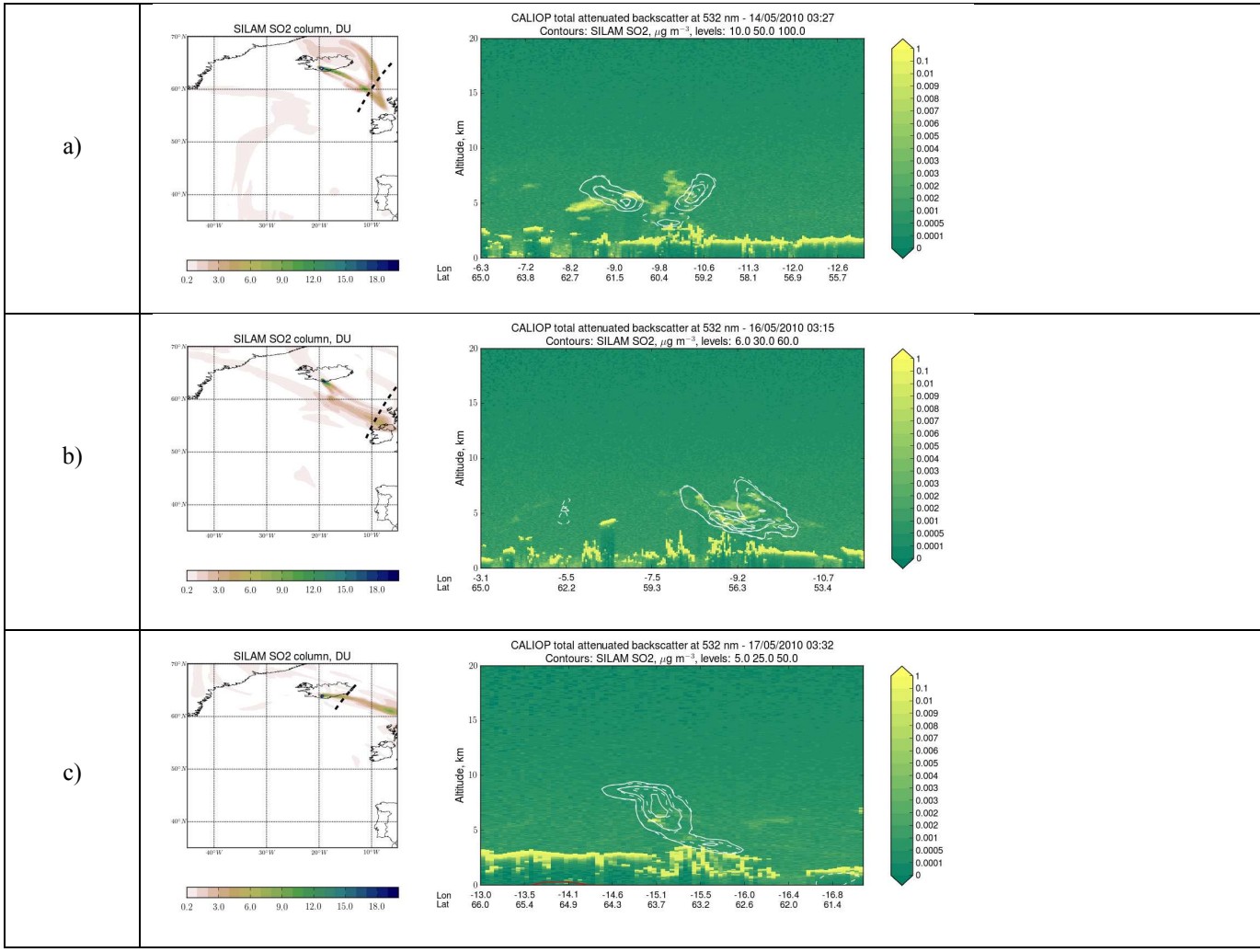

623

**Figure A1. Comparison of simulated SO₂ concentration compared to CALIOP total backscatter at 532 nm on 14 (panel a), 16 (b)**
**and 17 (c) May, 2010. The inversion with only total column retrievals is shown in dashed contours. The contour levels (µg m⁻³) are**
**10, 50 and 100 in panel a, 6, 30 and 60 in panel b and 5, 25 and 50 in panel c.**

627

## Appendix B: moments of products of correlated Gaussian random variables

Let $X$ and $Y$ be scalar random variables with means and variances $\mu_X$, $\mu_Y$, $\sigma_X^2$ and $\sigma_Y^2$. Then, it follows from the definitions for variance and covariance that

$$(12) \qquad \mathrm{Var}[XY] = \sigma_X^2 \sigma_Y^2 + \mu_X^2 \sigma_Y^2 + \mu_Y^2 \sigma_X^2 - 2\mu_X \mu_Y \mathrm{Cov}[X,Y] - \mathrm{Cov}[X,Y]^2 + \mathrm{Cov}[X^2,Y^2]$$

and

$$(13) \qquad \mathrm{Cov}[X,XY] = E[X^2]E[Y] + \mathrm{Cov}[X^2,Y] - E[X]E[XY] \; .$$

To expand $\mathrm{Cov}[X^2,Y^2]$ and $\mathrm{Cov}[X^2,Y]$ we assume that $X$ and $Y$ are normally distributed. We first define normalized auxiliary variables

$$(14) \qquad \tilde{X} = \frac{X - \mu_X}{\sigma_x}, \tilde{Y} = \frac{Y - \mu_Y}{\sigma_Y}$$

Then, by expressing $\tilde{Y}$ as

$$(15) \qquad \tilde{Y} = c\tilde{X} + \sqrt{1 - c^2}\,\tilde{Z}$$

where $c = \mathrm{Cov}[\tilde{X},\tilde{Y}]$ and $\tilde{Z} \sim \mathcal{N}(0,1)$ independent of $\tilde{X}$, it is simple to verify that

$$(16) \qquad \begin{aligned} \mathrm{Cov}[\tilde{X}^2,\tilde{Y}^2] &= 2c^2 \\ \mathrm{Cov}[\tilde{X}^2,\tilde{Y}] &= 0. \end{aligned}$$

For the original random variables $X$ and $Y$, we find by substituting (14) into the definition, expanding the terms, and using identities (16) that

$$(17) \qquad \mathrm{Cov}[X^2,Y^2] = 2\mathrm{Cov}[X,Y]^2 + 4\mu_X \mu_Y \mathrm{Cov}[X,Y]$$

and

$$(18) \qquad \mathrm{Cov}[X^2,Y] = 2\mu_X \mathrm{Cov}[X,Y] \; .$$

Formulas (6) and (7) now follow by combining Eqs. (17) and (18) with (12) and (13).

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

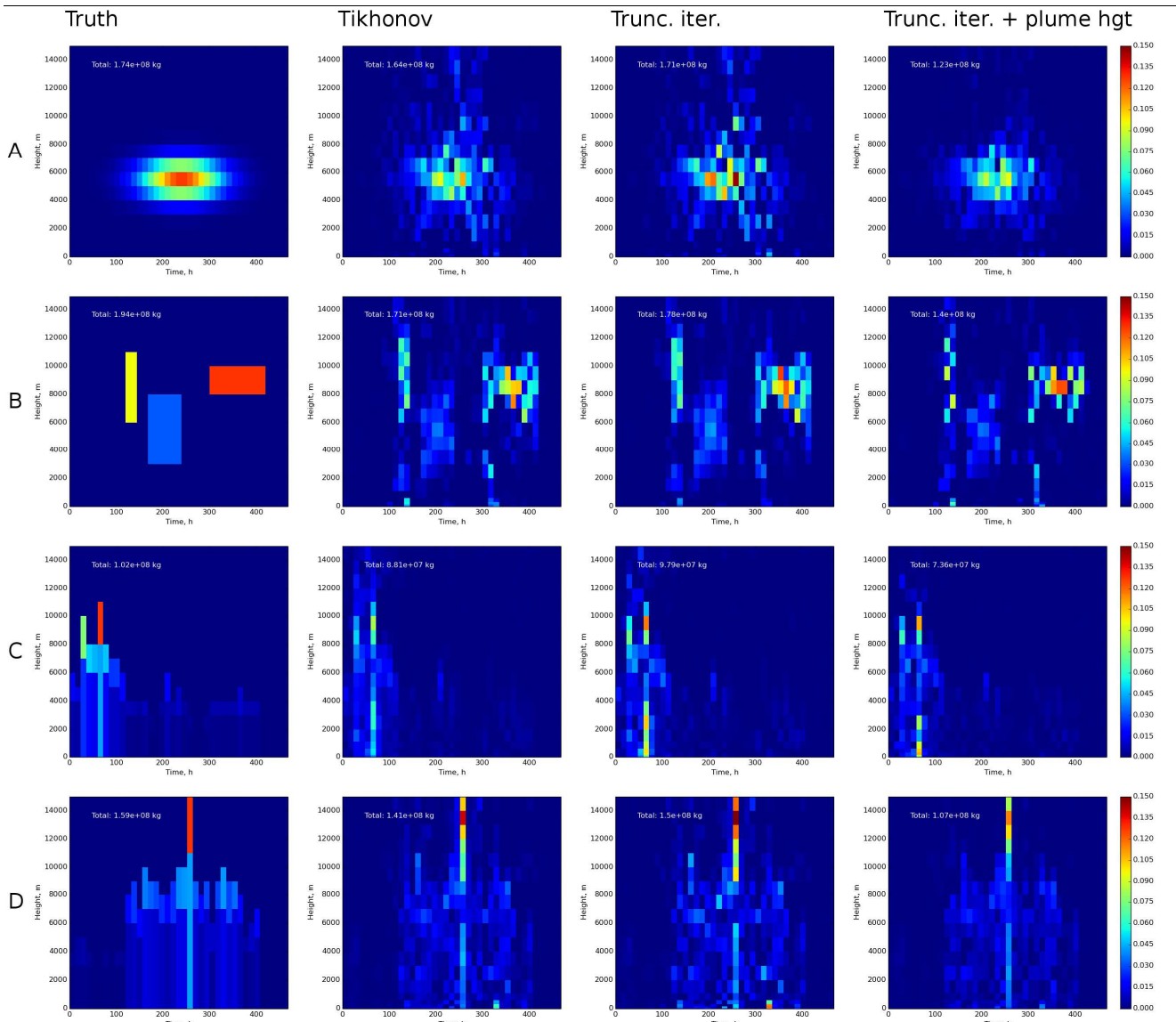

**Figure 1. Estimated emission flux (kg m$^{-1}$ s$^{-1}$) in source term inversions with simulated data. True source terms for the four cases (A to D) are shown in the left column. The remaining columns show the inversion results using Tikhonov regularisation, using truncated iteration with total column data, and using truncated iteration with total column and plume height data.**

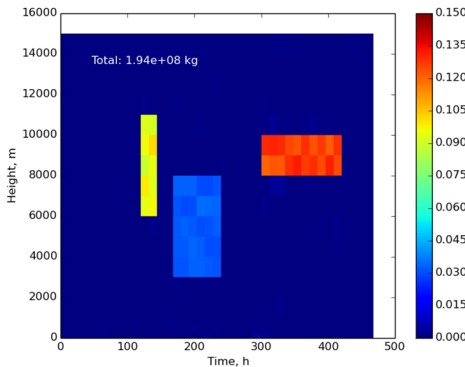

**Figure 2. Estimated emission flux with synthetic data: inversion results for the case B in Figure 1 assuming a perfect forward**
**model.**
**Table 1. Bias and RMSE with respect to the true source term (case A…D) in experiments with synthetic data with assimilation of**
**total column (TC) and total column and plume height (TC+CM). Values are shown for both optimal regularisation (regularisation**
**parameter or iteration number with the lowest RMSE) and for the regularisation chosen from L-curve. Relative bias is defined as**
**the difference between estimated and true total emission divided by the true total emission.**

| Case | | Tikhonov regularisation | | | | Truncated iteration | | | |
|---|---|---|---|---|---|---|---|---|---|
| | | RMSE | | Relative bias | | RMSE | | Relative bias | |
| | | Optimal | L-curve | Optimal | L-curve | Optimal | L-curve | Optimal | L-curve |
| A | TC | 48.0 | 48.0 | -5 % | -5 % | 45.2 | 51.2 | -3 % | -2 % |
| | TC+CM | 39.8 | 39.8 | -19 % | -19 % | 36.5 | 36.7 | -17 % | -17 % |
| B | TC | 65.1 | 65.6 | -8 % | -12 % | 61.4 | 61.9 | -8 % | -8 % |
| | TC+CM | 59.3 | 60.2 | -18 % | -23 % | 56.9 | 58.4 | -18 % | -17 % |
| C | TC | 21.1 | 21.1 | -13 % | -13 % | 20.6 | 21.9 | -8 % | -4 % |
| | TC+CM | 18.5 | 18.6 | -20 % | -24 % | 17.8 | 18.1 | -17 % | -17 % |
| D | TC | 32.4 | 33.6 | -15 % | -11 % | 31.1 | 38.0 | -8 % | -6 % |
| | TC+CM | 29.3 | 29.5 | -27 % | -24 % | 27.3 | 28.0 | -24 % | -21 % |



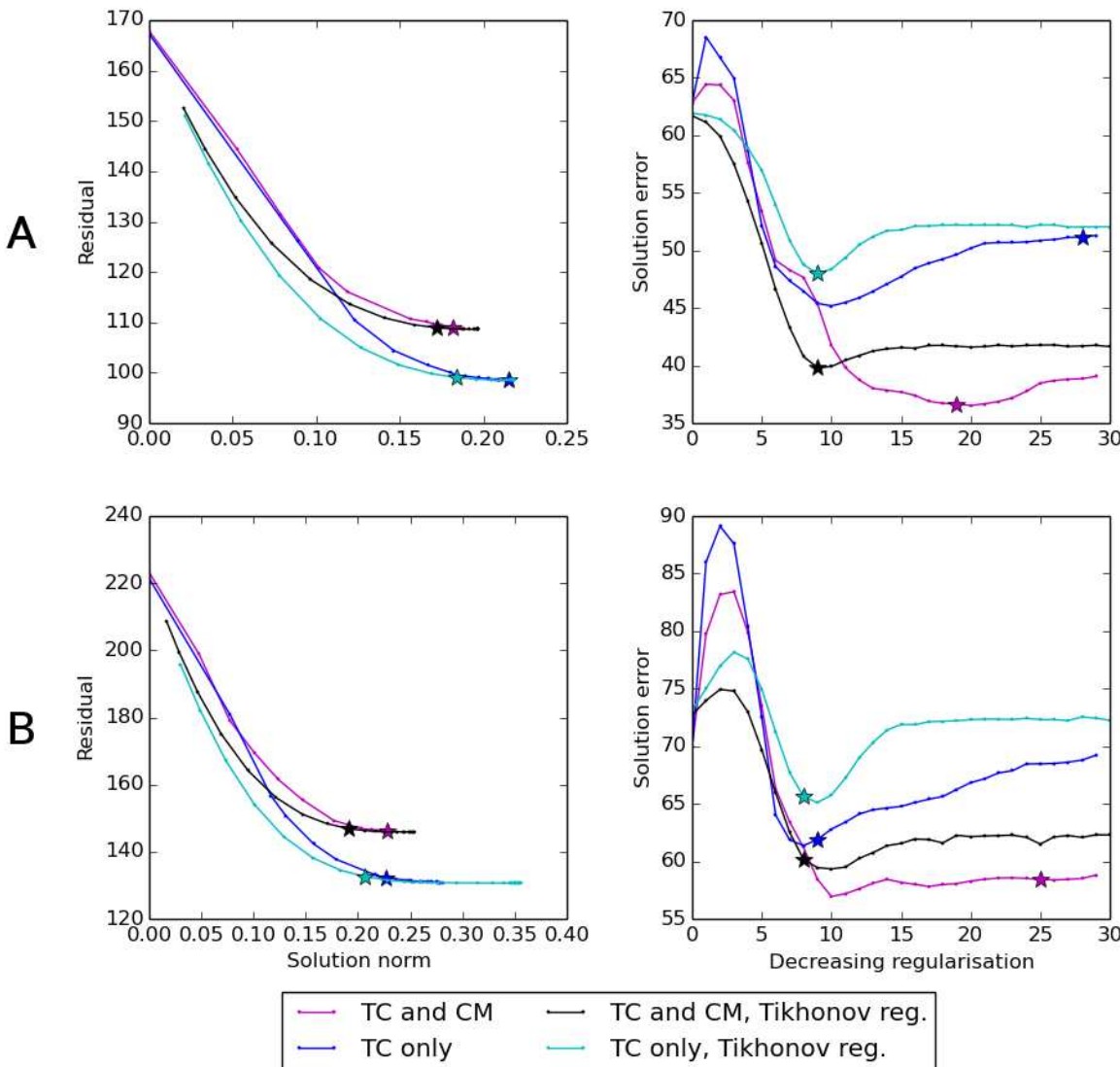


Figure 3. L-curve (left) and RMS error (right) for inversions with simulated data for cases A and B in Figure 1. The iterate (for truncated iteration) or the regularisation parameter (for Tikhonov regularisation) chosen from the L-curve is marked with a star.

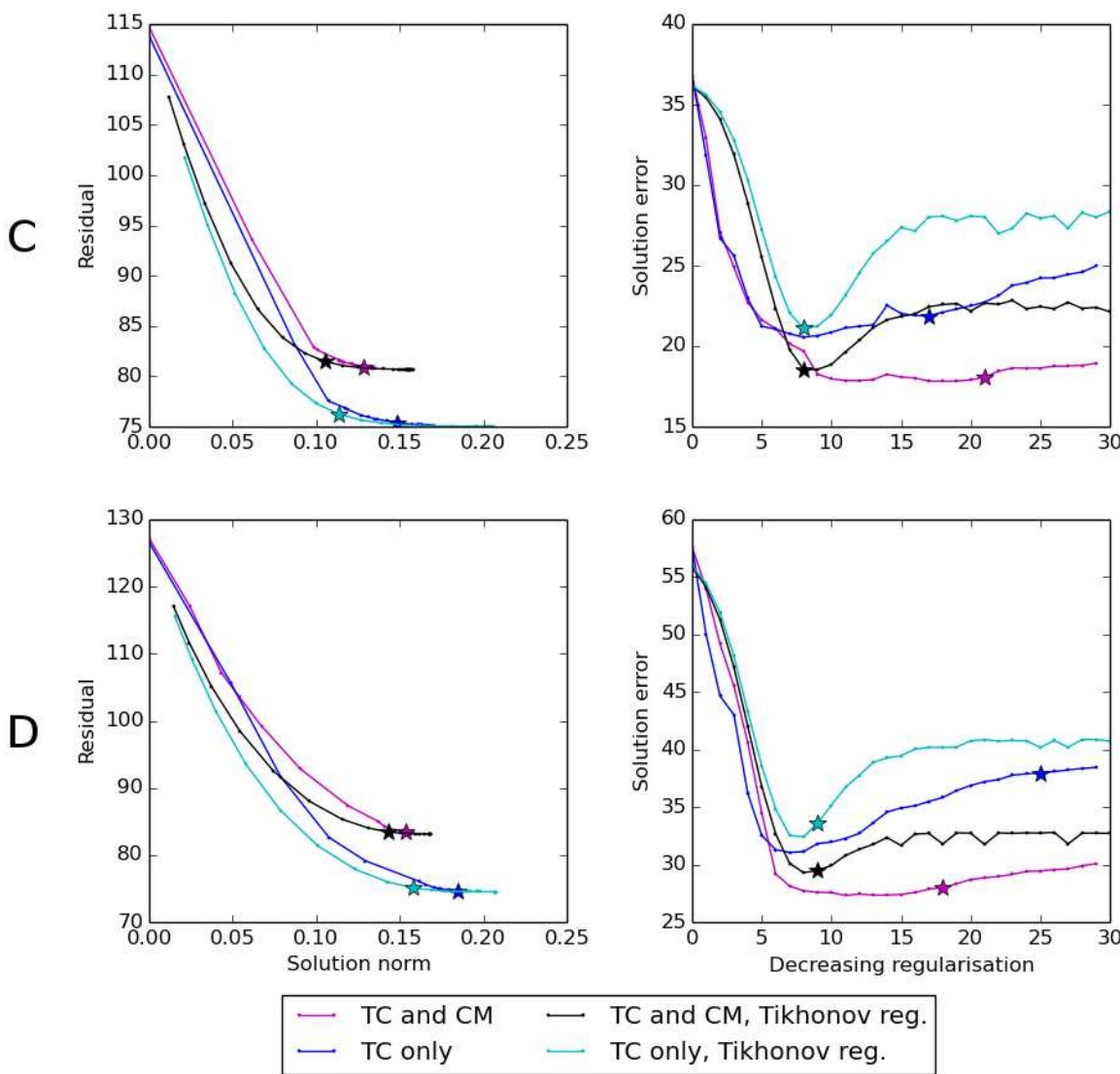

**Figure 4. L-curve (left) and RMS error (right) for inversions with simulated data for cases C and D in Figure 1. The iterate (for**
**truncated iteration) or regularisation parameter (for Tikhonov regularisation) chosen from the L-curve is marked with a star.**

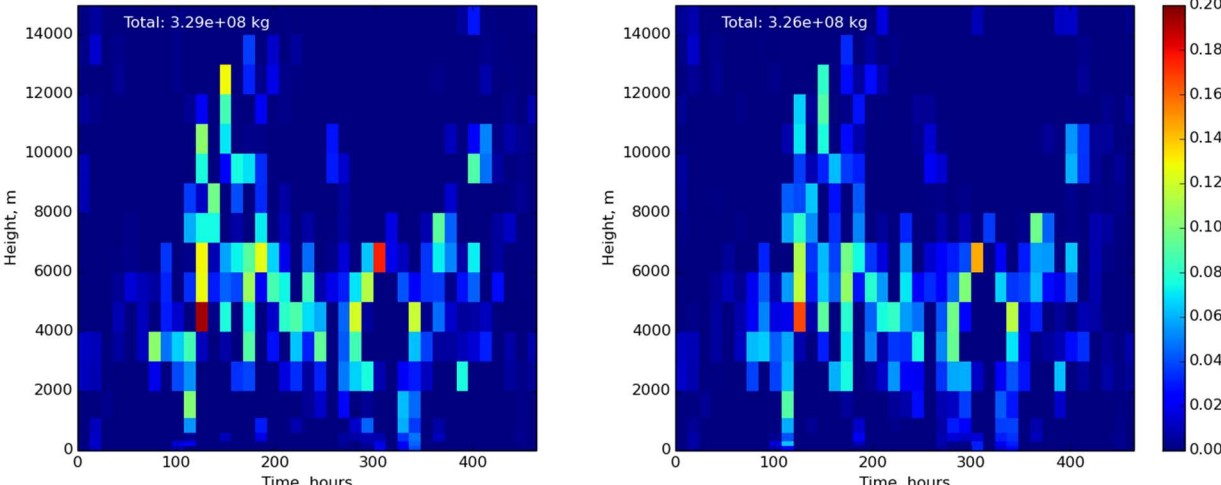

**Figure 5. Inversion results with real observations: emission flux (kg m⁻¹ s⁻¹) obtained using 4D-Var (left) and by evaluating the**
**sensitivity matrix (right). The inversions are based on total column observations.**

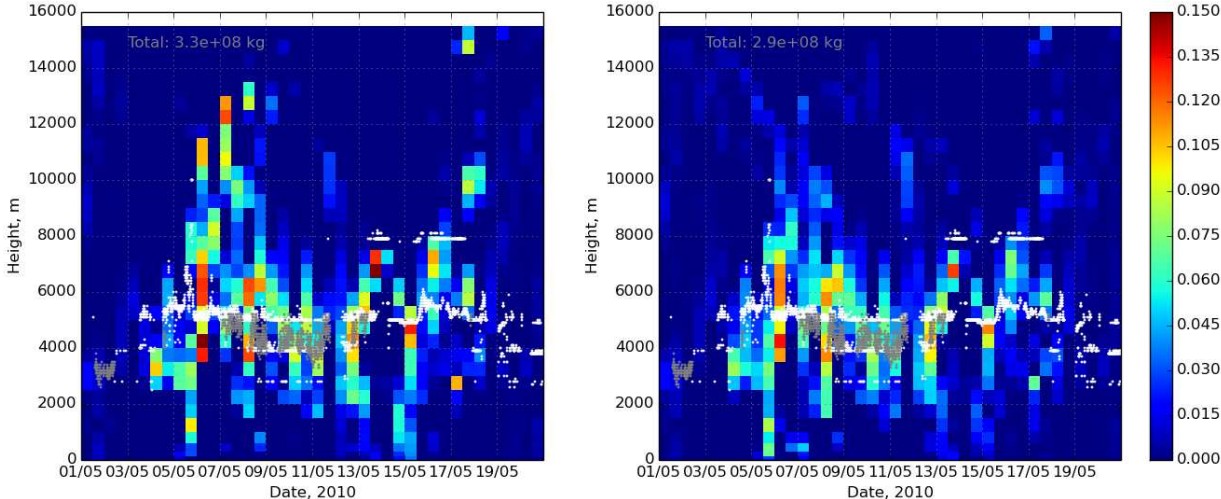



**Figure 6. Inversion results for Eyjafjallajökull. Left: emission flux (kg m⁻¹ s⁻¹) with assimilation of column mass only. Right:**
**assimilation of column mass and plume height with full observation error covariance matrix. White dots denote plume height**
**observations by radar, grey dots denote plume height observations with a camera.**

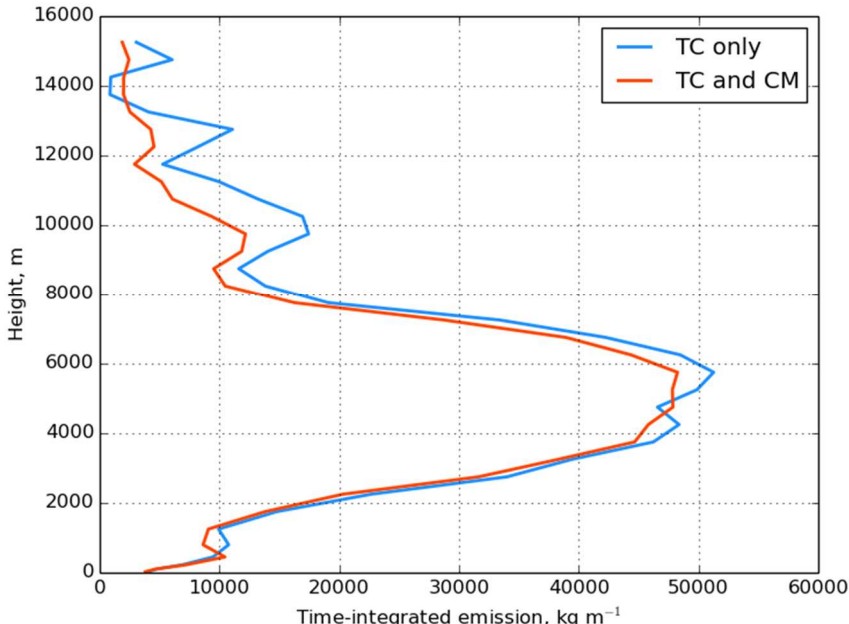

**Figure 7. Time-integrated emission of SO₂ (kg m⁻¹) during the simulated period as function of height (m) for the source term**
**inversions with (red) and without (blue) plume height assimilation.**


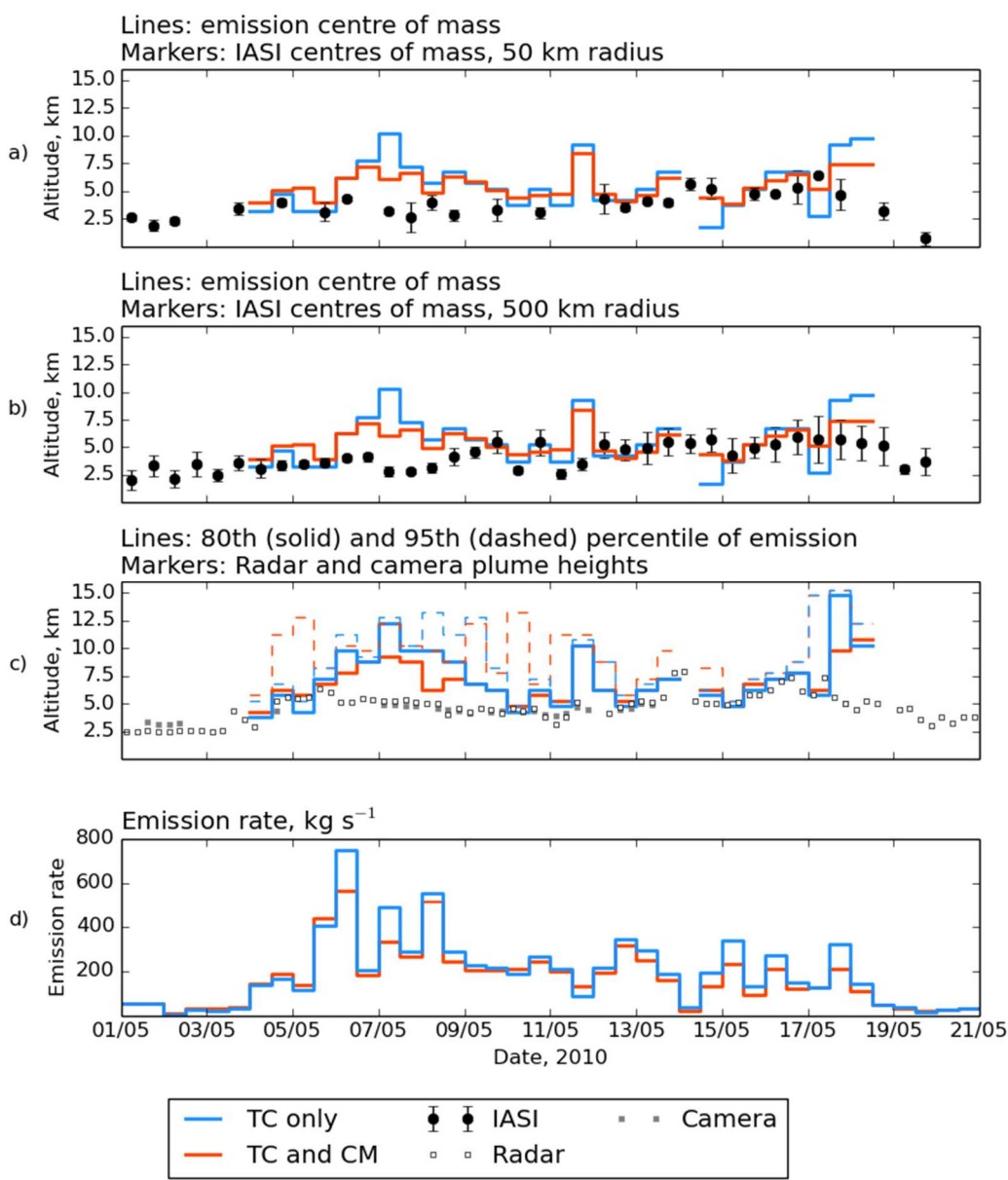


Figure 8. Inversion results for Eyjafjallajökull. Panels a and b: centre of mass of SO₂ injection and the average IASI plume height within 50 and 500 km from the volcano; panel c: 95th and 80th percentiles of SO₂ injection and the plume top altitudes observed by radar and camera; panel d: estimated emission rate (kg s⁻¹). Inversions using only total column retrievals are plotted in blue; inversions using total column and plume height retrievals are plotted in red. Fully correlated errors are assumed for evaluating the error bars for IASI data. The data with retrieval error estimate larger than 5 km are not included. The radar and camera observations are averaged to time steps of 6 hours. The centres of mass and percentiles of the inversion results are evaluated for the 12 hour steps emitting at least 1% of the total emission. All altitudes are above sea level.


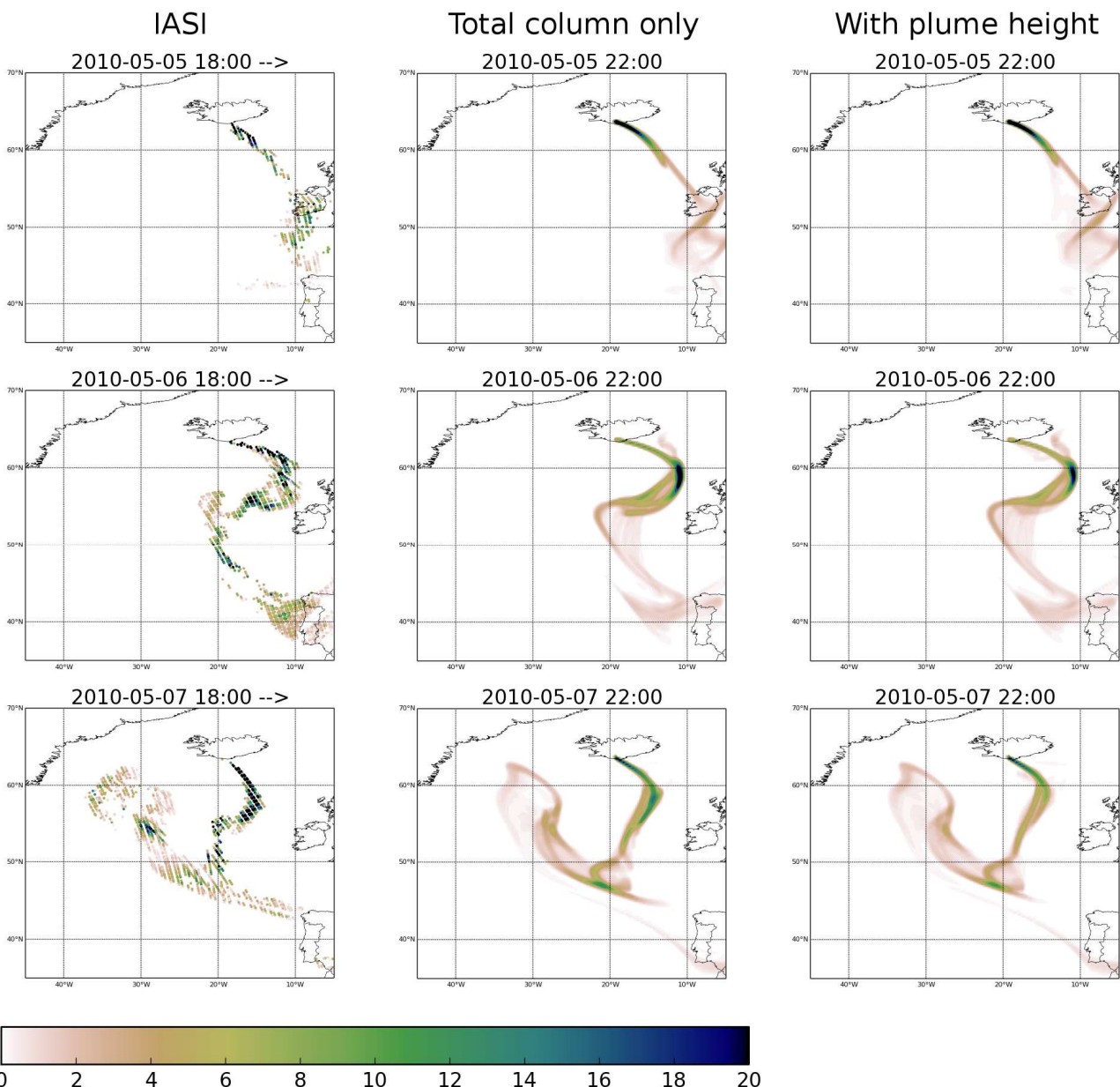

**Figure 9. SO$_2$ column loading (DU) for the IASI column retrievals (left column), for the a posteriori simulation with assimilation of**
**total column only (middle) and with assimilation of total column and plume height retrievals. Results for 5, 6 and 7th May, 2010**
**are shown in the rows from top to bottom. The evening overpasses are shown for IASI, the model fields are valid at 22 UTC.**




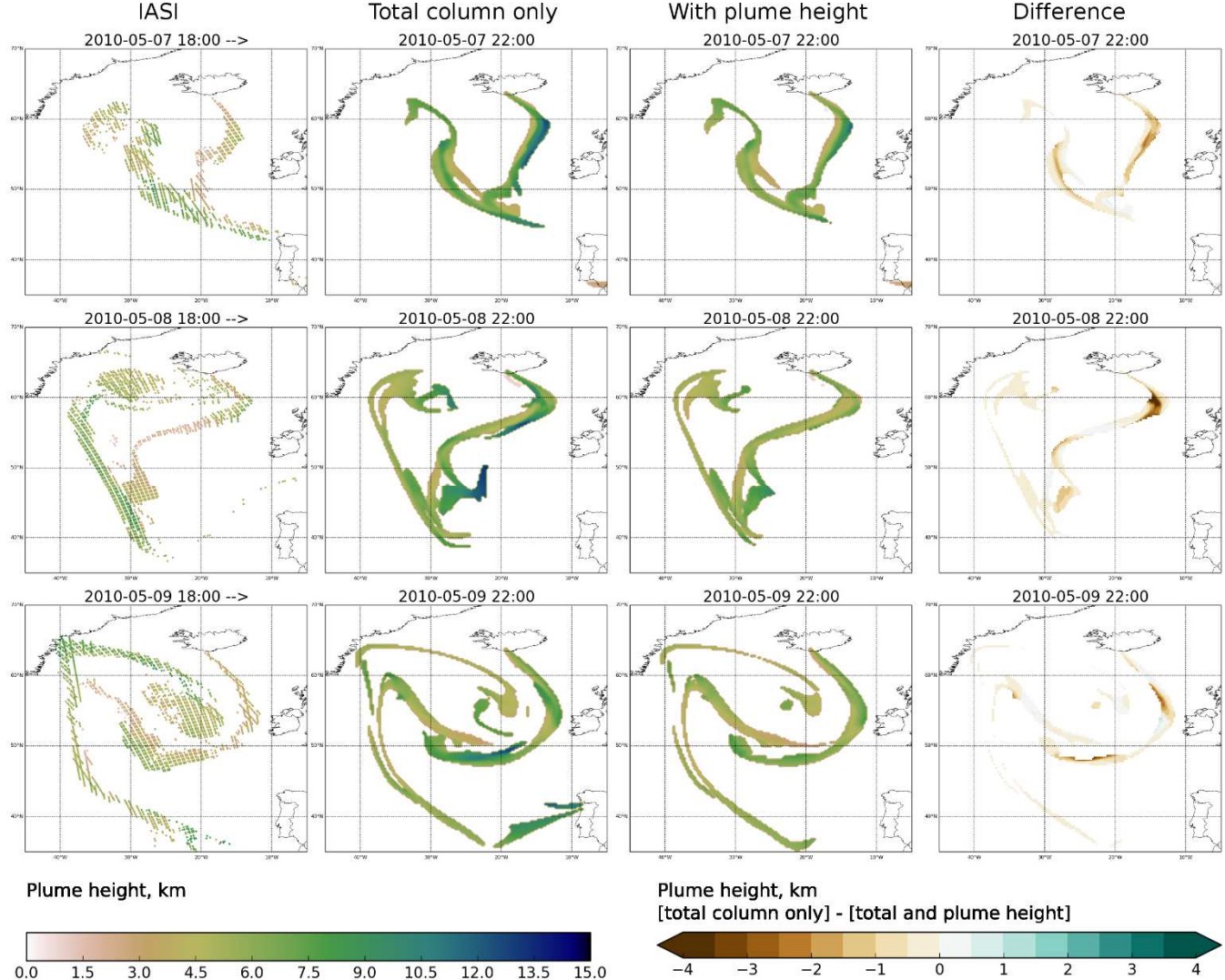


Figure 10. Retrieved SO$_2$ plume height (km, left column) and the simulated plume height (as centre of mass) without and with assimilation of plume height retrievals for 7-9 (top to bottom row) May, 2010. The difference (without plume height – with plume height) of the simulations is shown in the rightmost column.






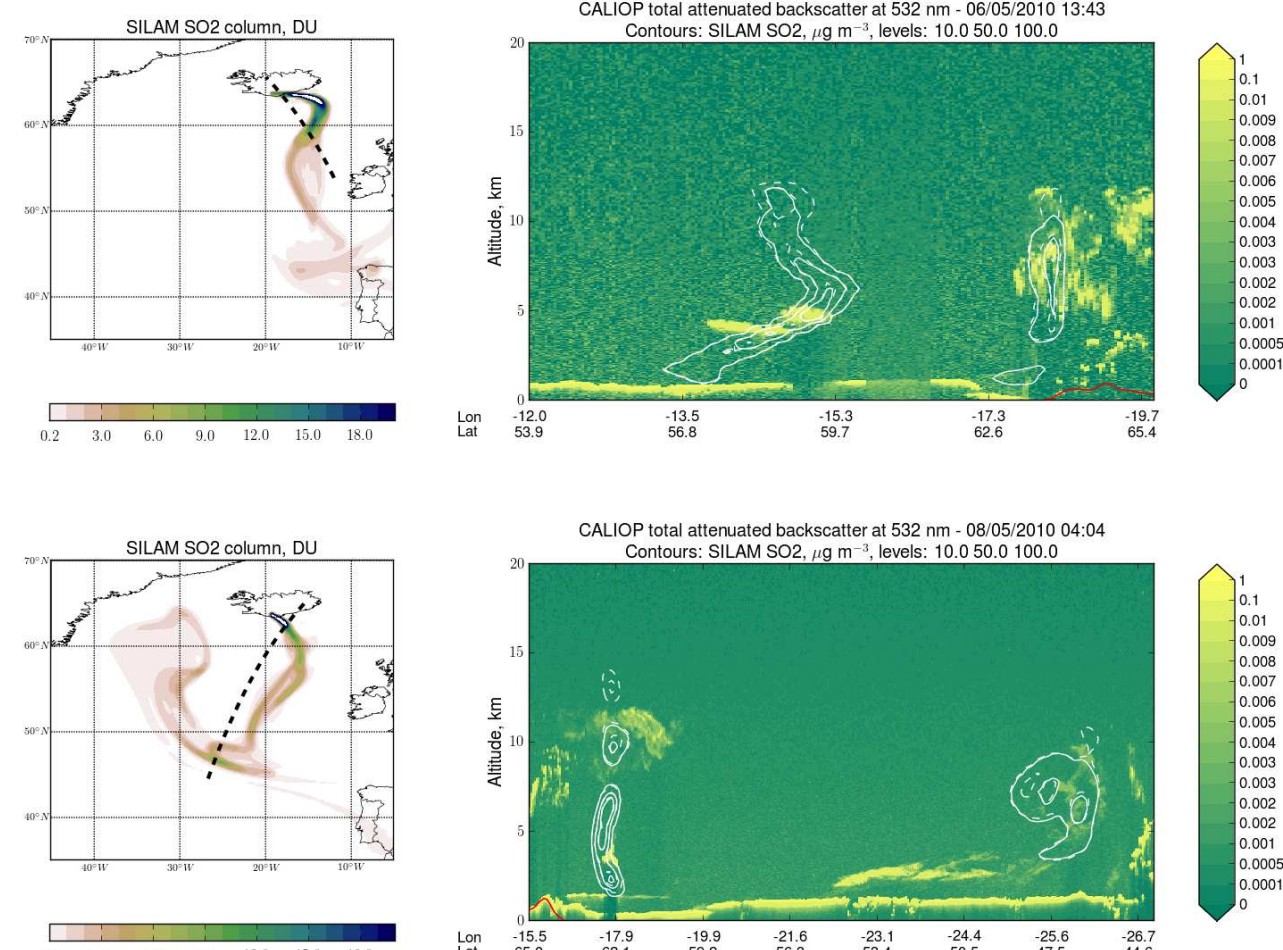



**Figure 11. Comparison of simulated SO₂ with CALIOP data for 14 UTC on 6 May (top) and 04 UTC on 8 May, 2010 (bottom).**
**Left: the simulated SO₂ total column (DU, with assimilation of both total column and plume height) with the CALIPSO track**
**plotted with dashed line. Right: CALIOP total attenuated backscatter at 532 nm with the simulated SO₂ concentration represented**
**by contours. The solid contours correspond to assimilation of both total column and plume height, the dashed contours correspond**
**to assimilation of total column only. The contour levels are 10, 50 and 100 μg m⁻³.**

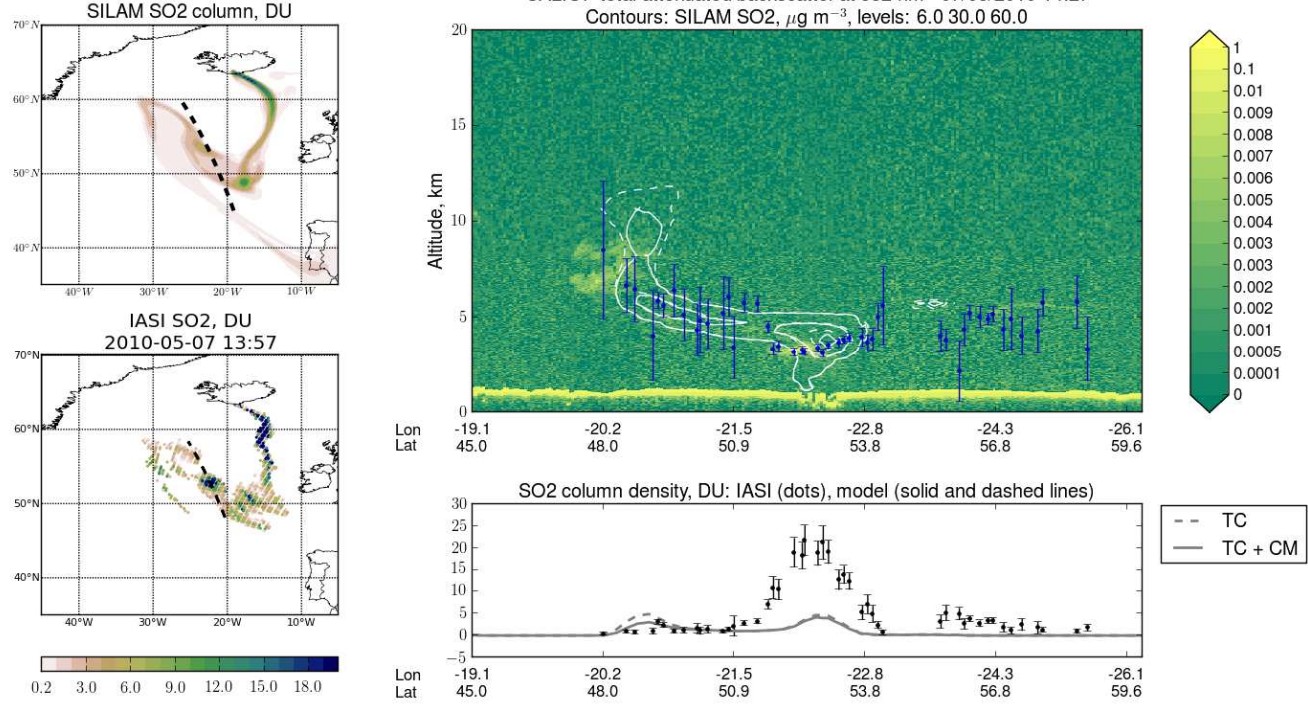

**Figure 12. CALIOP total attenuated backscatter, simulated SO₂ concentration (contour levels indicated on the figure title) and**
**collocated IASI plume height retrievals at ~14 UTC on May 7, 2010. The solid lines and contours correspond to inversion using**
**total column and plume height retrievals, dashed lines and contours correspond to inversion using total column retrievals only.**
**The modelled and retrieved column densities are shown in maps on the left and as a 1D plot along the CALIOP track on the**
**bottom. The full CALIOP track segment is marked in the map of simulated SO₂ columns (top-left), the track segment where the**
**collocated IASI data are extracted is shown in the map of retrieved SO₂ columns (bottom-left). The model SO₂ columns shown in**
**the map are from the inversion using both total column and plume height retrievals.**

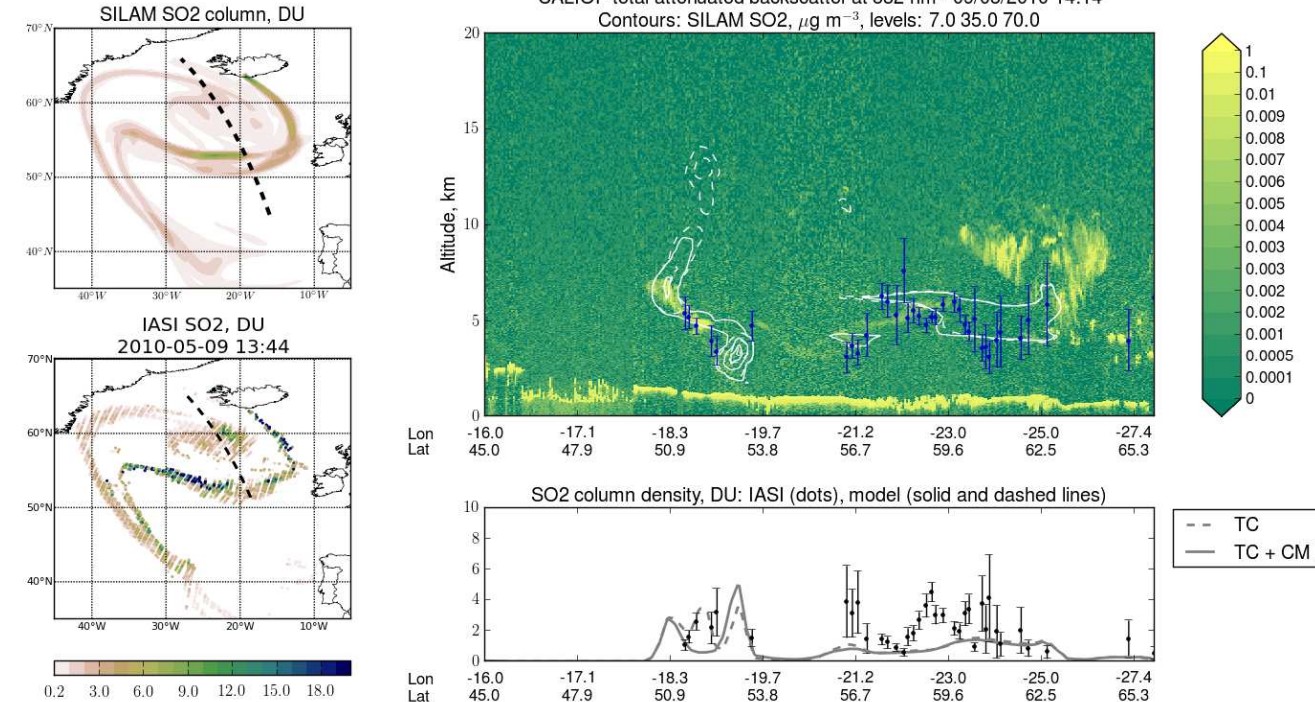


**Figure 13. As Figure 12 but for May 9, 2010.**

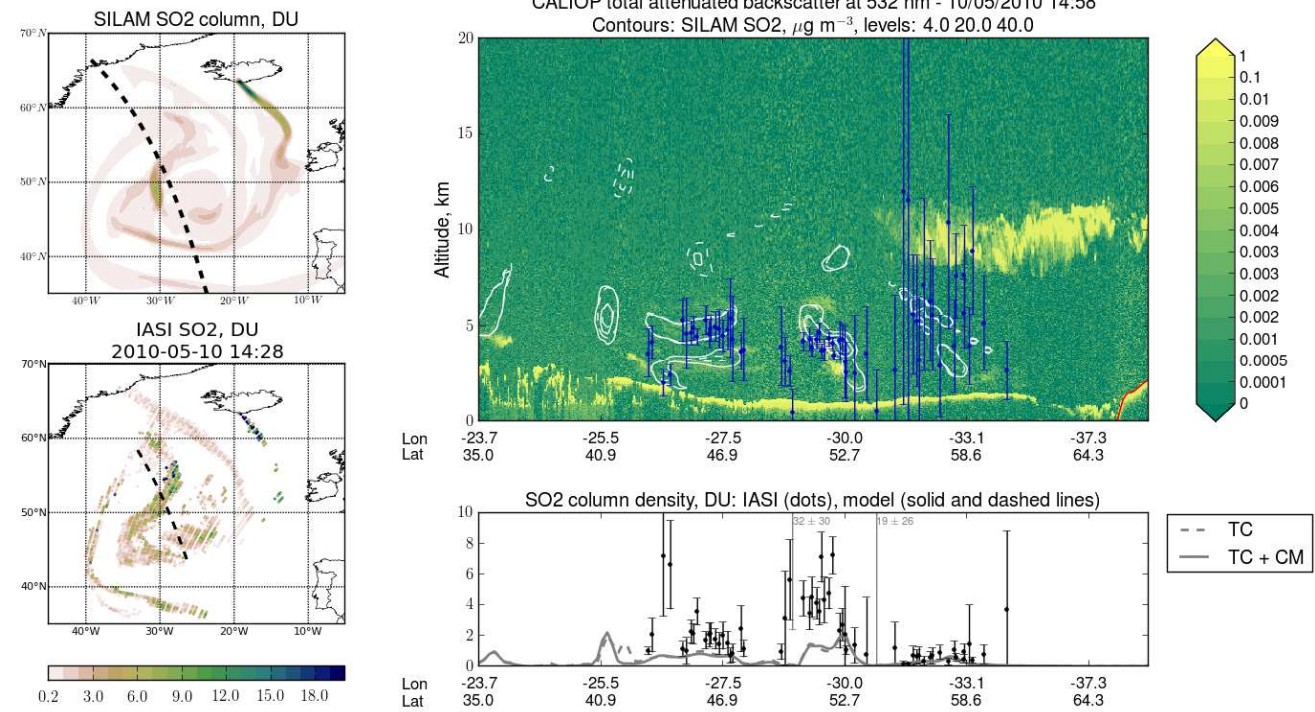


**Figure 14. As Figure 12 but for May 10, 2010. In the 1D column density plot below the CALIOP curtain, two IASI data points with**
**values 32±30 and 19±26 DU are outside the plot range.**