# Peer review of "Variational assimilation of IASI SO2 plume height and total-column retrievals in the 2010 eruption of Eyjafjallajökull using the SILAM v5.3 chemistry transport model"

_Geoscientific Model Development, 2016_

## Referee Comment (RC1) · Anonymous Referee #1 · 2 Nov 2016

Review of

Variational assimilation of IASI SO2 plume height and total-column retrievals in the 2010 eruption of Eyjafjallajökull using the SILAM v5.3 chemistry transport model.

by Julius Vira et al.

Overview:

The paper describes the assimilation of SO2 total column and plume height retrievals from IASI with the SILAM 4D-VAR system in order to infer the vertical and temporal

variability of the volcanic emissions.

First, the authors present an observations operator for the assimilation of plume height retrievals, which are available in addition to the retrieved total column of volcanic SO2 obtained from IASI. This is an interesting aspect because the retrieved plume height is a single value representing the centre of the plume whereas observed volcanic SO2 plume often occur in complex profiles and even multiple layers.

Second, the authors present an effective regularisation approach of the 4D-VAR problem, which allows the inversion of the source term without a-priori information. This is also an interesting aspect because most retrieval studies rely on an a-priori estimate of the source term.

The new method is tested with an artificial data set and with a real-world application of the Eyjafjallajökull eruption 1-20 May 2010.

General remarks

The paper addresses two important aspects of the assimilation of IASI SO2 retrievals, which are very valuable to the scientific community. However, the paper misses a convincing evaluation of the results.

The evaluation should be carried much more thoroughly. Independent data should be used or - if these are not available - the improvement of plume forecast in a near-real time scenarios where future observations are not available for the source term estimate could be estimated. The quantitative comparison with ash plume observations is not satisfactory. Likewise a better comparison with emission flux ( temporal variability and plume height) with the different estimates form the literature should be included.

In the current version of the manuscript, it remains unclear what the benefit of the assimilation of the plume height is. Given the overall uncertainty and judging from the pictures, it seems that the plume height assimilation does not lead to an improvement. If this is the case it should be mentioned more clearly.

The presented 4D-VAR approach does not take into account correctly the error statistics of the assimilating model as no model error co-variances are considered. This seems a simplification which should be better justified. Also, it seems that the SILAM model did not consider a chemical loss for SO2, which – if this was the case – would be an unnecessary simplification of the model.

The paper would benefit from a more detailed and thorough description of the experiments as well as a more detailed discussion and conclusion section.

Specific remarks

L 17: Clarify "vertical centre of mass" and "first moment"

L 20: Mention the relation of regularization with the a-priori estimate here.

L 50: Discuss the issue of the "single value" plume height retrieval and the observed complex SO2 profiles and the resulting challenges for the assimilation.

L 64: Clarify the differences between the "inversion type studies" using Lagrangian models, the 4D-VAR approach for the assimilation of only concentrations and the inclusion of the emission term in these 4D-VAR systems.

L 91: Say what the data set entails: TC and plume heights retrievals.

L 99: What is the effective plume height?

L 103: If this is the ash plume height - why the ash plume height?

L 155: What are inversion experiments. Please clarify the use of the terms "data assimilation" and "inversion" throughout the paper (see L64).

L 122: The "inversion" with simulated observations is discussed before the Eyjafjallajökull case. So please also mention them here before.

L 128: Is the conversion of SO2 to SO4 considered in the study ?

L 140: Please discuss that the standard 4D-VAR approach would include the model

error in form of the background error covariance matrix.

L 146: Spell out L-BFGS-B

L 156: Define also y and m_ij

L 160: Provide the formulae for centre of mass and 1st moment of mass

L 175: It is not clear if R also contains the error of the plume height retrieval. Is this error also provided and used?

L 204: This is not surprising given that the model/background error is ignored in equation (2)

L 209: Having only a diagonal model error covariance ignores the fact that the model advects the tracer.

L 219: Provide reference or explanation for the Tikhonov regularisation

L 252: Please describe the setting of the synthetic experiment better. What was period, region and meteorological data ? Is only the emission term synthetic ? What are the synthetic observations?

L 315: What is meant by "overall need"? Why is there an assumption about the source term? I thought regularizations is introduced to avoid an prior assumption for the source term? From which of the above are these generalisations deduced?

L 316: In which of the experiments there was no model error? Please clarify. Perhaps a table of all the synthetic experiments would be useful.

L 320: Say exactly which type of experiment you refer to.

L 322: Motivate this choice of the 9th and 13th iterate better. I thought that the L-curve needs to be examined for every case specifically.

L 324: Over plotting the observed ash plume height in Figure 7 is not really instructive. You should try to plot the time series of the averaged emission centre from the two

inversions together with the retrieved plume height (see Figure 13) in the vicinity of the volcano. This would show how much the plume height observations in the whole domain constrain the injection profile over the volcano. The observed ash plume height, which is basically the lower border of the ash plume, is less instructive.

Consider showing he difference between the two inversions in Figure 7b.

L 328: Stating the differences between the two inversions is not enough (see my general remark). Please say which of the two inversions is better. If this is no possible, say more clearly that this is the case.

L 337: Please discuss in more detail, why the total emission change despite that fact that the assimilated total columns are the same.

L 341: A plot of the difference with the base case would be clearer.

L 343: Check language "... as spread as ..."

L 344: Again, it needs to be shown, that the results shown in Figure 10 are better or worse than the ones from Figure 7 before any conclusion can be drawn.

L 349: In line 337 you say it is about 10%. Why is this the case? Were the observations errors different for the assimilated TC in the two cases? The TC and the plume height retrievals are not independent,

L 350: Please provide more evidence and discussion.

L 355: I find it actually quite interesting that the additional assimilation of the plume height has so little influence.

L 357: Again, this is no proof that one inversion is better.

L 366: The differences between GOME-2 and IASI plume height retrievals should be discussed in more detail. The explanation is too short.

L 366 and L 370: Compare your results for both the injection height and the SO2 mass

with the results published in the literature, for example Boichu et al. 2013, Flemming and Inness, 2013. This would be an important result and should be mentioned in the conclusion or abstract.

L 376: Which other studies? Provide references.

L 380: This is all a bit unspecific. Please quantify the identified noise and how this could relate to the results from the 2010 case study.

L 390: 1000 days wall clock time or simulated days? How long does it need on a typical high performance computer architecture? What are the options for the parallelism of the application.

L 400: It think it is fair to say that the assimilation of the plume height only had a small influence on the results. Also, the paper provides not enough evidence that one option is better than the other.

---

## Referee Comment (RC2) · Anonymous Referee #2 · 4 Dec 2016

**Overview**

This paper describes the development of an observation operator for the simultaneous assimilation of the OMI retrievals of total SO2 columns and "plume height" (or more correctly, height of centre of mass) associated with volcanic eruptions; the implementation of this observation operator into an existing variational assimilation system; its testing and set-up using synthetic observations; and its application to the Eyjafjallajökull eruption 1-20 May 2010.

As explained by the first reviewer already, this research is timely and of high interest. While a large amount of work was invested in the development of the observation operator, the improvement due to plume height assimilation seems marginal and requires much more thorough evaluation. I will not focus on the inversion results for the real-life case and the corresponding discussion, first because this was already addressed in the first review and second because I have serious concerns about the assimilation approach itself. These concerns may be due to misunderstandings on my side, reflecting a lack of clarity and completeness in the manuscript. In such a case the manuscript requires major revisions in sections 3 and 4. If on the other side the concerns raised here can not be alleviated, the assimilation approach is wrong (which could explain the limited success to improve the estimation of plume height) and I would recommend a serious re-examination of the assimilation algorithm before re-running all the inversion experiments - i.e. most probably a withdrawal of the current manuscript.

**Absence of a priori term in the cost function**

I believe that the authors used a deeply flawed implementation of the variational assimilation method. Canonical expressions of the cost function (see e.g. Talagrand, 1997) decompose it into a first term $\frac{1}{2} (\mathbf{x}-\mathbf{x}^b)^T \mathbf{B}^{-1} (\mathbf{x}- \mathbf{x}^b)$ measuring the distance between the a priori (i.e. background) and optimized model state, and a second term measuring the mismatch between model state and observations. The success of this approach relies on a proper estimation and balance between the corresponding background error covariance matrix (**B**) and observations error covariance matrix (**R**). Even though systematic model errors (biases) are often neglected in such variational assimilation systems, the a priori errors can not be ignored.

As best as I can guess, the authors decided that the cost of the a priori information could be neglected because the optimized parameter **f** is taken as zero during the first forward model integration (although this is not clearly stated in section 3). The authors probably chose as initial background model state $\mathbf{x}_0^b$ (i.e. initial condition for the first iteration) the output at the end of the previous assimilation window (note this too should have been clearly stated in section 3). The corresponding first guess is thus that the volcanic eruption suddenly ended at the beginning of the current assimilation window, with $SO_2$ abundance simply dissipating due to advection and

decreasing due to photochemistry. Such a priori information is quite far from the truth during the eruption, and the associated background error must be quite large. In any case, the background term $\frac{1}{2}(\mathbf{x}-\mathbf{x}^b)^T\mathbf{B}^{-1}(\mathbf{x}-\mathbf{x}^b)$ can clearly not be zero – except during the first iteration of the first assimilation window.

I suspect that the addition of a regularization term (section 3.4) was considered as an attempt to make up for the absence of a background term. I note that the regularization term is fundamentally different because it depends directly on the model parameter while the background term depends on the model parameter through the model state. In any case the regularization is abandoned in favour of a truncated iteration, which I only see as a way to prevent the system from wandering too far from its first guess despite the absence of the corresponding term in the cost function.

There are two clues that the absence of an a priori term in the cost function prevented the system from behaving correctly:

- The authors had to supplement the observation error with a model error (line 209: $\mathbf{R} = \mathbf{R}_{obs} + \mathbf{R}_{model}$). This makes no sense because model error should be considered in model (gridded) space and be applied to the model state. Its dimension excepted, this matrix $\mathbf{R}_{model}$ is exactly the first estimation which one could use to account for the impact of model errors on a priori error and evaluate $\mathbf{B}$. Indeed the background error covariance matrix is often approximated as diagonal, with values set arbitrarily from educated guesses of the model error (Errera and Ménard, 2012).

- The blue lines on the right plots in figures 3 and 4 shows that after an "optimal" number of iterations, the RMS error using synthetic observations increases again with the iteration number. One would expect from a well-behaved iterative assimilation system that once it has reached convergence, the number of iterations does not change the error of its output.

This leads us to the most basic question: does this system converge towards a stable solution? A revised paper should show the evolution of $J/n$ (cost function divided by number of observations) as a function of the iteration number, for a variety of (synthetic or real) observations and at different dates. This also provides an opportunity to argue for the validity of the current assimilation approach: in the current experiments, compute (a posteriori) the background cost $J_B = \frac{1}{2}(\mathbf{x}-\mathbf{x}^b)^T\mathbf{B}^{-1}(\mathbf{x}-\mathbf{x}^b)$ , using a diagonal $\mathbf{B}$ with values no smaller than those chosen for $\mathbf{R}_{model}$ . Does $J_B$ remain much smaller than the currently minimized cost $J_R$ as the number of iterations increase? Does the total cost $J_B + J_R$ converge towards a minimum value or does it start increasing after an "optimal" number of iterations?

**Other important remarks**

The submitted text requires many other clarifications, but it is not worth listing these in detail as long as the fundamental point raised above is not addressed. Yet some points are serious enough to require immediate action in a revised manuscript:

1. There is a systematic confusion between "plume height" and (height of) "centre of mass". Both quantities seem interchangeable, so I would expect that this was described in a previous paper about the retrieval scheme. This should still be explained with proper references in section 2.2, with much more attention paid to the exact term used throughout the text (including title and abstract). Even if this was published already, consider adding a line plot in section 2.2 to compare the retrieved centre of mass with the plume height observed by radar and cameras (i.e. white dots and barely visible grey dots in figure 7). Note also that such a comparison directly above Eyjafjallajökull would still not be convincing for the downstream plume, where a multi-layered distribution probably happened.

2. What is the width of the Gaussian distribution assumed by the retrieval algorithm? The observation operator does not take into account this Gaussian shape. We are in a case where the vertical resolution leading to the simulated observation is much, much finer than the "vertical resolution" of the observation itself so this looks like a major oversight - especially in a context where Averaging Kernels were not taken into account. In order to simulate the observations correctly, the observation operator should first fit the modelled profile with a Gaussian shape before applying equation (4). Of course this should also be included in the adjoint of the observation operator…

3. The sentence on lines 134-135 is extremely problematic as it seems to show a fundamental misunderstanding about assimilation theory:

   > "Finally, the vector **y** of observations is given by the possibly non-linear observation operator $\mathcal{H}$ as **y**=$\mathcal{H}$(x)+ε where ε denotes the observation error."

   $\mathcal{H}$(**x**) is the model state in observation space. **y**-$\mathcal{H}$(x) is the observation departure. Here ε does not denote only the observation error but all possible errors, including the model and representatitvity errors. Most importantly, **y** is not "given" by $\mathcal{H}$, it is simulated by $\mathcal{H}$ ! This awful confusion continues in equation (3) which does not provide **y** but the (total component of) $\mathcal{H}$(**x**).

4. It appears from lines 307-312 that the assimilation algorithm can use both the algebraic solution with explicit computation of **HM** matrix, and the 4D-Var approach. This should be clearly explained in section 3.1. I assume that Vira and Sofiev (2012) provided all necessary details about this 4D-Var implementation and its adjoint model (e.g., was it generated automatically or manually? Is it as detailed as the forward model? How was it verified?). If that is the case, an additional reference and a few words may suffice. But if that was not the case, or if the 4D-Var implementation changed a lot (beyond the developments described in sections 3.2 and 3.3), then this should be fully described (since appropriate for the GMD journal).

**References**

Errera, Q. and Ménard, R., Technical Note: Spectral representation of spatial correlations in variational assimilation with grid point models and application to the Belgian Assimilation System for Chemical Observations (BASCOE), Atmos. Chem. Phys., 12, 10015-10031, 2012.

Talagrand, O.: Assimilation of observations, an introduction, J. Meteorol. Soc. Jpn, 277, 191–209, 1997.

---

## Author Comment (AC1) · 13 Dec 2016

**Author comment regarding the assimilation method**

We thank the referee for the comments. The referee raises concerns regarding the validity of our inversion method. However, we are convinced that this is indeed a misunderstanding, caused by differing conventions within the fields of data assimilation and inverse modelling. In the following, we aim to address the question regarding validity of the assimilation method; we will address the specific comments (1-4) in the final response provided along with submission of the revised manuscript.

In our inversion, contrary to numerical weather prediction, the observations are assimilated in a single window covering all 21 days. We realise that this aspect may not have been spelled out clearly enough in the manuscript, as it is only mentioned implicitly in section 3.1 (line 133: "state vector x is taken collectively for all time steps"). This important point will be stated upfront in the revised manuscript.

Little $SO_2$ was observed at the beginning of the assimilation window, which coincided with inactive phase of the eruption. It is therefore reasonable to assume that the $SO_2$ concentrations within the assimilation window depended only on the emission flux $\mathbf{f}$, given as a function of time and altitude. In this case, the cost function does not include a term for the background state $\mathbf{x_b}$, and instead, the possible a priori constraints are given for the emission $\mathbf{f}$:

(1)
$$\mathcal{J}(\mathbf{f}) = \frac{1}{2}(\mathbf{y} - \mathcal{H}(\mathbf{x}))^T \mathbf{R}^{-1}(\mathbf{y} - \mathcal{H}(\mathbf{x})) + \mathcal{T}(\mathbf{f})$$

where $\mathbf{x} = \mathcal{M}(\mathbf{f})$ includes the model state on all time steps of the assimilation window, $\mathcal{H}$, $\mathbf{y}$, and $\mathbf{R}$ denote the observation operator, observation data, and observation error covariance matrix, and $\mathcal{T}(\mathbf{f})$ is a penalty functional. Most of the existing studies on inverting volcanic emissions have used a cost function similar to Eq. (1) with various forms of $\mathcal{T}(\mathbf{f})$, including an a priori emission source with a prescribed prior error covariance matrix (Eckhardt et al. (2008) and the subsequent works using the FLEXPART model), or Tikhonov regularisation with first or second order smoothness constraints (Boichu et al., 2013; Lu et al., 2016). Since there is little prior information on smoothness of volcanic emissions, we chose the arguably simplest option of zeroth order Tikhonov regularisation (Eq. 9 in the manuscript) as the point of reference.

A limitation of inversions based on Eq. (1) is that the model is taken as a strong constraint i.e., model errors developing within the assimilation window are not included in the cost function. Since the model errors are not negligible, their effect on the residual $\mathbf{y} - \mathcal{H}(\mathbf{x})$ needs to be included into the observation error covariance matrix $\mathbf{R}$. Also this aspect has been recognized by previous flux inversion studies for both volcanic (Seibert et al., 2011; Stohl et al., 2011) and other emissions (eg. Bergamaschi et al. (2005); Bocquet, (2012)).

It would be possible to arrange the inversion into successive assimilation windows as done by Elbern et al. (2007) in context of air quality forecasting. In this case, the initial state should probably be included in the control vector; however, this would not remove the assumption of negligible model errors within the assimilation window. This approach is so far untested for volcanic emissions – Flemming and Inness (2013) estimated the emission fluxes in a separate step.

The reviewer suggests that the truncated iteration is only a way to "prevent the system from wandering too far", with which we respectfully disagree. Truncated iteration (often simply called iterative regularisation) has been shown to be a computationally efficient method for regularising large-scale inverse problems, and it is discussed extensively in textbooks (e.g. Hansen, 2010; Kaipio and Somersalo, 2006) in addition to the papers (Calvetti et al., 2002; Fleming, 1990; Kilmer and O'Leary, 2001) cited in the manuscript.

The behaviour of RMSE as a function of iteration number is therefore not an error, but fully consistent with what is expected for many iterative methods when applied to an ill-posed problem. As seen in the left panels of Figs. 3 and 4, the residual indeed decreases as the iteration proceeds. The eventual growth of RMSE, however, is an indication of overfitting the solution to noise. The noise will eventually contaminate the converged solution, whereas the early iterates are controlled by more robust low-frequency features of the data. Truncating the iteration prevents the overfitting and results in a more accurate solution than that obtained by allowing the iteration to converge.

Although the self-regularising properties of iterative methods are rarely exploited in data assimilation, in our experiments the truncated L-BFGS-B iteration yielded solutions that were, depending on the point of truncation, similar or better in RMSE than those given by optimally tuned Tikhonov regularisation. With Tikhonov regularisation (Eq. 9), the iteration converges, although with weak regularisation (small $\alpha^2$), the convergence becomes very slow.

Due to the reasons explained above, the tests with $J_B$ and $J_R$ as suggested in the review are not actually possible. However, we can use the $\chi^2$ condition to evaluate the consistency of the $\mathbf{R}$ matrix. For a well-specified inversion, linear regression theory implies (e.g. Tarantola, 2005) that the expectation of the unregularised cost function (Eq. 2 in the manuscript) at the minimum is

$$(2) \qquad\qquad E(J(\mathbf{f}_{opt})) = \frac{n-p}{2}$$

where $n$ is number of observations and $p$ is number of estimated parameters. For the Eyjafjalljökull inversion with assimilation of total column and plume height (centre of mass) retrievals, $p \sim 1400$, $n \sim 1.4 \cdot 10^6$ and $J(\mathbf{f}_{opt}) \sim 6 \cdot 10^4$ with $\mathbf{f}_{opt}$ given by the L-curve criterion. The apparent discrepancy is explained by the large proportion of zero observations with only small contribution to the total cost function. If the cost function is evaluated using only observations corresponding to positive $SO_2$ detections, we have $n_{y>0} \sim 10^5$ and $J_{y>0}(\mathbf{f}_{opt}) \sim 5 \cdot 10^4$. Even if Eq. (2) does not strictly hold for subsets of observations, this suggests that our $\mathbf{R}$ is reasonably specified, especially given that the assumed values for $\mathbf{R_{model}}$ are at best a crude approximation for the actual model uncertainty.

**References**

Bergamaschi, P., Krol, M., Dentener, F., Vermeulen, A., Meinhardt, F., Graul, R., Ramonet, M., Peters, W., Dlugokencky, E.J., 2005. Inverse modelling of national and European CH4 emissions using the atmospheric zoom model TM5. Atmos. Chem. Phys. 5, 2431–2460. doi:10.5194/acpd-5-1007-2005

Bocquet, M., 2012. Parameter-field estimation for atmospheric dispersion: application to the Chernobyl accident using 4D-Var. Q. J. R. Meteorol. Soc. 138, 664–681. doi:10.1002/qj.961

Boichu, M., Menut, L., Khvorostyanov, D., Clarisse, L., Clerbaux, C., Turquety, S., Coheur, P.-F., 2013. Inverting for volcanic SO2 flux at high temporal resolution using spaceborne plume imagery and chemistry-transport modelling: the 2010 Eyjafjallajökull eruption case study. Atmos. Chem. Phys. 13, 8569–8584. doi:10.5194/acp-13-8569-2013

Calvetti, D., Lewis, B., Reichel, L., 2002. GMRES, L-curves, and discrete ill-posed problems. BIT Numer. Math. 42, 44–65.

Eckhardt, S., Prata, A.J., Seibert, P., Stebel, K., Stohl, A., 2008. Estimation of the vertical profile of sulfur dioxide injection into the atmosphere by a volcanic eruption using satellite column measurements and

inverse transport modeling. Atmos. Chem. Phys. 8, 3881–3897. doi:10.5194/acpd-8-3761-2008

Elbern, H., Strunk, A., Schmidt, H., Talagrand, O., 2007. Emission rate and chemical state estimation by 4-dimensional variational inversion. Atmos. Chem. Phys. 7, 3749–3769. doi:10.5194/acpd-7-1725-2007

Fleming, H.E., 1990. Equivalence of regularization and truncated iteration in the solution of Ill-posed image reconstruction problems. Linear Algebra Appl. 130, 133–150. doi:10.1016/0024-3795(90)90210-4

Flemming, J., Inness, A., 2013. Volcanic sulfur dioxide plume forecasts based on UV-satellite retrievals for the 2011 Grímsvötn and the 2010 Eyjafjallajökull eruption. J. Geophys. Res. Atmos. 118. doi:10.1002/jgrd.50753

Hansen, P.C., 2010. Discrete Inverse Problems: Insight and Algorithms, Fundamentals of Algorithms. Society for Industrial and Applied Mathematics.

Kaipio, J., Somersalo, E., 2006. Statistical and Computational Inverse Problems, Applied Mathematical Sciences. Springer New York.

Kilmer, M.E., O'Leary, D.P., 2001. Choosing Regularization Parameters in Iterative Methods for Ill-Posed Problems. SIAM J. Matrix Anal. Appl. 22, 1204–1221. doi:10.1137/S0895479899345960

Lu, S., Lin, H.X., Heemink, A.W., Fu, G., Segers, A.J., 2016. Estimation of Volcanic Ash Emissions Using Trajectory-Based 4D-Var Data Assimilation. Mon. Weather Rev. 144, 575–589. doi:10.1175/MWR-D-15-0194.1

Seibert, P., Kristiansen, N.I., Richter, A., Eckhardt, S., Prata, A.J., Stohl, A., 2011. Uncertainties in the inverse modelling of sulphur dioxide eruption profiles. Geomatics, Nat. Hazards Risk 2, 201–216. doi:10.1080/19475705.2011.590533

Stohl, A., Prata, A.J., Eckhardt, S., Clarisse, L., Durant, A., Henne, S., Kristiansen, N.I., Minikin, A., Schumann, U., Seibert, P., Stebel, K., Thomas, H.E., Thorsteinsson, T., Tørseth, K., Weinzierl, B., 2011. Determination of time- and height-resolved volcanic ash emissions and their use for quantitative ash dispersion modeling: the 2010 Eyjafjallajökull eruption. Atmos. Chem. Phys. 11, 4333–4351. doi:10.5194/acp-11-4333-2011

Tarantola, A., 2005. Inverse problem theory and methods for model parameter estimation. Society of Industrial and Applied Mathematics.

---

## Author Response (AR1)

**Response to reviewer comments**

J. Vira and co-authors

We thank both referees for the extensive reviews. In the following we will address the comments of reviewer #1 and the specific comments of reviewer #2. The general comment of reviewer #2 has been addressed in our response on 13 December 2016.

The main changes in the revised manuscript is extending experiments with synthetic data in Section 4 to cover assimilation of simulated plume height retrievals, and presenting a comparison between the simulated SO2 profiles and the lidar data from the CALIOP instrument. These major revisions are discussed in more detail in our response to reviewer #1.

Due to the volume of new material, we have removed the discussions of two minor items present in the original submission. The omissions are Figure 5 and the related discussion (additional experiments with randomly generated source terms), and Figure 10 and the related discussion (inversion with a simplified observation error covariance matrix).

The figures for Eyjafjallajökull have been rearranged as part of the comparisons for total columns are now presented together with the CALIOP data. Figures comparing the simulated total columns and plume heights with the IASI data for all days are included in a supplement.

The manuscript with changes highlighted is attached to this pdf.

*The reviewer comments are presented with a blue italic font.*

**Response to comments by reviewer #1**

We thank the referee for a thorough and detailed review. In order to address the main concerns raised by the referee, we have introduced the following major revisions to the manuscript:

1.  The experiments with synthetic observations are extended to cover the case with assimilation of simulated plume height retrievals. The results are consistent with those obtained for the Eyjafjallajökull eruption. Assimilation of plume height retrievals results in a more accurate source term (in the sense of RMS error), although the difference is small for some of the assumed source terms. However, the experiment also confirms that assimilation of plume heights may have negative impact on the estimated total emission.
2.  The results for Eyjafjallajökull are evaluated using profile data from the CALIOP instrument, and a detailed comparisons between the estimated injection height, the retrieved SO2 plume heights, and the radar observations of plume top are presented. The comparisons show consistently that the results with and without assimilation of plume heights are largely similar but the vertical distribution of the emissions especially on 6-9 May is improved by assimilation of plume height retrievals. For the other times, the injection height is mostly consistent with the data regardless whether the plume heights are assimilated or not.

Ideally the results for total columns should be compared with independent satellite data. However, as noted in earlier studies about Eyjafjallajökull (Boichu et al., 2013; Flemming and Inness, 2013), this is unlikely to lead into useful conclusions due to the large differences between satellite products. For this reason, we have instead focused on the vertical distribution which we consider to be the most interesting of aspect of the present study.

*In the current version of the manuscript, it remains unclear what the benefit of the assimilation of the plume height is. Given the overall uncertainty and judging from the pictures, it seems that the plume height assimilation does not lead to an improvement. If this is the case it should be mentioned more clearly.*

We believe that the revised manuscript gives a more accurate and detailed picture of impact of assimilating the plume height. However, we hesitate to declare one inversion better than other, because drawing such a conclusion would require three-dimensional observational data that do not exist. The possible benefits of assimilating the plume height in addition to total column cannot be evaluated based on their impact on the simulated total columns or the total emission, because these quantities are fit optimally when only total column data are assimilated.

Based on the experiments with synthetic data and the comparison to vertically resolved observations for Eyjafjallajökull, it seems reasonable to conclude that assimilating the plume height retrieval has a positive impact on the vertical distribution, even if the effect is small when taken over the whole eruption. Similarly, the experiments indicate that the impact on total columns and total emission is negative. Whether one of these effects outweighs the other depends on the application.

*The presented 4D-VAR approach does not take into account correctly the error statistics of the assimilating model as no model error co-variances are considered. This seems a simplification which should be better justified.*

We agree that the treatment of model errors is not ideal, however, as pointed out in the initial response to reviewer #2, the problem is challenging and we are not aware of any inversion study that would have addressed this aspect without simplifications.

Inclusion of the model errors into the R-matrix is necessary because the inversion is performed in a single, long assimilation window. As discussed in the revised manuscript, it would be in principle possible to set up the inversion as a sequence of shorter assimilation windows as done by Elbern et al. (2007) in context of air quality forecasting. However, when the primary interest is to estimate an emission source, this approach becomes more complicated because of two opposing requirements. The first requirement is that the assimilation window needs to be short enough so that the model errors arising within the assimilation window are not significant – this is the basic assumption of the currently used strong-constraint assimilation methods. The second requirement is that the assimilation window should be long enough to effectively constrain the emission. Shorter assimilation windows may be useful for constraining constant or slowly varying emissions, but such assumptions would not hold for volcanic emissions.

So far, the majority of studies on inverse modelling of atmospheric emissions have emphasized the second aspect and used long assimilation windows. This includes the inversions for volcanic emissions (Boichu et al., 2013; Kristiansen et al., 2010; Lu et al., 2016; Stohl et al., 2011), where the inversion is done for the entire eruption at once, but also inversions for other trace gas emissions (e.g. Meirink et al. 2008; Müller and Stavrakou, 2005) or for inverse modelling of accidental radioactive releases (e.g. Winiarek et al., 2014). An advantage of the long assimilation window without explicit model errors is that the no unphysical sources or sinks appear due to the assimilation, and the mass budget of the simulation remains closed. While this does not imply that the long-window approach is superior to the other options, a comparison of its advantages and disadvantages is outside the scope of the current paper.

*Also, it seems that the SILAM model did not consider a chemical loss for SO2, which – if this was the case – would be an unnecessary simplification of the model.*

As noted on L76 in Section 2.1 (Dispersion model), the model does include chemical removal of SO2.

*L 17: Clarify "vertical centre of mass" and "first moment"*

Done.

*L 20: Mention the relation of regularization with the a-priori estimate here.*

Done.

*L 50: Discuss the issue of the "single value" plume height retrieval and the observed complex SO2 profiles and the resulting challenges for the assimilation.*

A discussion has been added onwards of L65.

*L 64: Clarify the differences between the "inversion type studies" using Lagrangian models, the 4D-VAR approach for the assimilation of only concentrations and the inclusion of the emission term in these 4D-VAR systems.*

Additional discussion has been added on lines 70-76.

*L 91: Say what the data set entails: TC and plume heights retrievals*

"The algorithm and the dataset are explained in more detail by Carboni et al. (2012)" has been changed into "The algorithm and the IASI SO2 dataset (column amount and altitude) are explained in more detail by Carboni et al. (2012)".

The effective altitude is the altitude of the mean of the Gaussian distribution that fits the measurements. We called it "effective" because we assume one SO2 layer with a Gaussian profile. To clarify we have changed the text: "The scheme determines the column amount and effective altitude of the SO2 plume" is changed into: "The scheme determines the column amount and the altitude (mean of Gaussian distribution) of the SO2 plume".

*L 103: If this is the ash plume height - why the ash plume height?*

This is the altitude of the SO2 plume, not the ash plume; to avoid misunderstanding we have changed the text:

"The altitude retrieved for the Eyjafjallajökull eruption plume"

into

"The SO2 altitude retrieved for the Eyjafjallajökull eruption".

*L 155: What are inversion experiments. Please clarify the use of the terms "data assimilation" and "inversion" throughout the paper (see L64).*

The present study is best described as a (source term) inversion similarly to the studies cited in our response to the comment regarding the 4D-Var method. We nevertheless use the word "assimilate" in reference to the action of an observation operator (as in "to assimilate plume height retrievals"), since this is both intuitive and consistent with existing literature on inverse modelling.

*L 122: The "inversion" with simulated observations is discussed before the Eyjafjallajökull case. So please also mention them here before.*

Done.

*L 128: Is the conversion of SO2 to SO4 considered in the study?*

Yes, as indicated above. A remark has been added also here.

*L 140: Please discuss that the standard 4D-VAR approach would include the model error in form of the background error covariance matrix.*

We have included a discussion on model errors and possible ways to handle them.

*L 146: Spell out L-BFGS-B*

Done.

*L 156: Define also y and m_ij*

Done.

*L 160: Provide the formulae for centre of mass and 1st moment of mass*

Done.

*L 175: It is not clear if R also contains the error of the plume height retrieval. Is this error also provided and used?*

The plume height retrieval error is provided and used. The corresponding text has been added.

*L 204: This is not surprising given that the model/background error is ignored in equation (2)*

*L 209: Having only a diagonal model error covariance ignores the fact that the model advects the tracer.*

We refer to our response above regarding the 4D-Var formulation.

*L 219: Provide reference or explanation for the Tikhonov regularisation*

References have been added.

*L 252: Please describe the setting of the synthetic experiment better. What was period, region and meteorological data? Is only the emission term synthetic? What are the synthetic observations?*

The section has been rewritten for more clarity. With synthetic observations we refer to simulated observations and observation errors. In addition, we have simplified the section by omitting the additional experiment (Fig. 5) with randomly generated source terms.

*L 315: What is meant by "overall need"? Why is there an assumption about the source term? I thought regularizations is introduced to avoid an prior assumption for the source term? From which of the above are these generalisations deduced?*

By the need for regularisation we meant that the sensitivity to under-regularisation varied between the assumed sources. However, we agree that this was formulated unclearly, and the paragraph has been rephrased.

*L 316: In which of the experiments there was no model error? Please clarify. Perhaps a table of all the synthetic experiments would be useful.*

We have clarified that model error was present in all experiments except for creating Fig. 2.

*L 320: Say exactly which type of experiment you refer to*

Done.

*L 322: Motivate this choice of the 9th and 13th iterate better. I thought that the L-curve needs to be examined for every case specifically.*

The iterates were chosen according to the algorithm described on lines 364-366 in the revised manuscript. We also examined the L-curves visually but kept the points chosen by the algorithm. The L-curves are provided in supplementary information.

*L 324: Over plotting the observed ash plume height in Figure 7 is not really instructive. You should try to plot the time series of the averaged emission centre from the two inversions together with the retrieved plume height (see Figure 13) in the vicinity of the volcano. This would show how much the plume height observations in the whole domain constrain the injection profile over the volcano. The observed ash plume height, which is basically the lower border of the ash plume, is less instructive. Consider showing he difference between the two inversions in Figure 7b.*

As stated in Arason et al. (2011), the radar time series actually represent the upper border of the ash plume. While we agree that comparing the radar data with the inversion results is somewhat ambiguous, we have decided to keep the comparison, since the radar data provide an independent dataset evaluating the estimated injection height. However, we have also added a chart showing the comparison of the emission centre of mass with plume height retrievals averaged within 50 and 500 km radiuses from the volcano. We hope that Fig. 8 also exposes the differences between the inversion results in Fig. 7.

*L 328: Stating the differences between the two inversions is not enough (see my general remark). Please say which of the two inversions is better. If this is no possible, say more clearly that this is the case.*

Please see our response to the general remark.

*L 337: Please discuss in more detail, why the total emission change despite that fact that the assimilated total columns are the same.*

We have added (L562 onwards) a discussion regarding effects of the plume height assimilation on the total emission.

The total emission changes because when only total columns are assimilated, the inversion has more degrees of freedom to match the observed horizontal distribution. When a vertical constraint is added, the simulated total columns must change unless the model describes the real transport perfectly, which is not a realistic assumption. To give a concrete example, in Fig. 13 in the revised manuscript, the TC-only run appears to fit better the total column data between 50° and 54° N, shown in the lower panel with the line plots. However, the comparison with IASI and CALIOP data shows that this corresponds to an unrealistic vertical distribution. As a consequence, the TC+CM run agrees worse with the total column, but is quite likely to agree better with the true, three-dimensional distribution.

The synthetic experiments indicate that the TC+CM inversions tend to have a low bias for total mass, which is consistent with the results for Eyjafjallajökull. The likely reason is the regularisation, which among different emissions with similar likelihood prefers the one with lowest squared sum. Whether a different cost function would avoid the negative effect on total emission is a question for a future study.

*L 341: A plot of the difference with the base case would be clearer.*

*L 343: Check language "as spread as"*

*L 344: Again, it needs to be shown, that the results shown in Figure 10 are better or worse than the ones from Figure 7 before any conclusion can be drawn*

Due to the volume of new material added, and since this aspect is peripheral to the main discussion, we have omitted Fig. 10 and the associated text from the revised manuscript.

*L 349: In line 337 you say it is about 10%. Why is this the case? Were the observations errors different for the assimilated TC in the two cases? The TC and the plume height retrievals are not independent*

We rewritten the text for better self-consistency. The same observation errors were used in both experiments. The correlation between TC and plume height retrieval errors is taken into account as indicated now more clearly in Section 3.2.

*L 350: Please provide more evidence and discussion.*

More evidence (especially Fig. 8) and discussion has been added.

*L 355:  I find it actually quite interesting that the additional assimilation of the plume height has so little influence.*

*L 357: Again, this is no proof that one inversion is better*

A more detailed discussion about the changes and similarities between the results has been added.

*L 366:  The differences between GOME-2 and IASI plume height retrievals should be discussed in more detail. The explanation is too short.*

We mention the GOME-2 retrievals, since they are one of the few published observations on SO2 plume height. As indicated, neither our data nor inverse modelling reproduces the GOME-2 plume height, but investigating the reasons for this would require a detailed comparison of the satellite products and the retrieval algorithms.

*L 366 and L 370: Compare your results for both the injection height and the SO2 mass with the results published in the literature, for example Boichu et al.  2013, Flemming and Inness, 2013.  This would be an important result and should be mentioned in the conclusion or abstract.*

Comparisons with Boichu et al. (2013), Flemming and Inness (2013) and Stohl et al. (2011)  have been added.

*L 376: Which other studies? Provide references.*

References (Boichu et al., 2013; Seibert et al., 2011) have been added.

*L 380:  This is all a bit unspecific. Please quantify the identified noise and how this could relate to the results from the 2010 case study.*

The section has been revised based on the new results using synthetic plume height observations in Section 4.

*L 390: 1000 days wall clock time or simulated days? How long does it need on a typical high performance computer architecture?  What are the options for the parallelism of the application.*

The 1000 days are simulated days. We have added a discussion about the required wall clock time and the options for parallelisation in each case.

*L 400: It think it is fair to say that the assimilation of the plume height only had a small influence on the results. Also, the paper provides not enough evidence that one option is better than the other.*

We agree that the difference is mostly minor. We have expanded the Conclusions section to cover both similarities and differences between the inversions as well as the impacts on both vertical distribution and total mass.

**Response to specific comments by reviewer #2**

This response extends our earlier response posted on 13 December 2016 and addresses the specific remarks by reviewer #2.

*The reviewer comments are presented with a blue italic font.*

*There is a systematic confusion between "plume height" and (height of) "centre of mass". Both quantities seem interchangeable, so I would expect that this was described in a previous paper about the retrieval scheme. This should still be explained with proper references in section 2.2, with much more attention paid to the exact term used throughout the text (including title and abstract). Even if this was published already, consider adding a line plot in section 2.2 to compare the retrieved centre of mass with the plume height observed by radar and cameras (i.e. white dots and barely visible grey dots in figure 7). Note also that such a comparison directly above Eyjafjallajökull would still not be convincing for the downstream plume, where a multi-layered distribution probably happened.*

The term "plume height retrieval" is used previous literature (Carboni et al., 2016, 2012; Rix et al., 2012), where it is interpreted as the midpoint of some parametric representation of the vertical profile. Since our observation operator does not assume a specific shape of the profile, it is more accurately described as an observation operator for the centre of mass (or rather, the vertical first moment of mass). However, since this definition is rather technical we prefer to use the term plume height retrieval except in cases where the distinction is essential.

We have added to the introduction a remark regarding the usage of "plume height" and "centre of mass" in the revised manuscript. The manuscript now uses "plume height" in all non-technical contexts.

The revised manuscript includes a comparison between the simulated emission, the IASI retrievals and the plume top observations, and also comparisons of the simulated SO2 concentration with the CALIOP profiles and collocated IASI retrievals.

*What is the width of the Gaussian distribution assumed by the retrieval algorithm? The observation operator does not take into account this Gaussian shape. We are in a case where the vertical resolution leading to the simulated observation is much, much finer than the "vertical resolution" of the observation itself so this looks like a major oversight - especially in a context where Averaging Kernels were not taken into account. In order to simulate the observations correctly, the observation operator should first fit the modelled profile with a Gaussian shape before applying equation (4). Of course this should also be included in the adjoint of the observation operator...*

The width of the Gaussian used in IASI forward model is 100 mb. The retrievals were also tested assuming different widths (50 and 10 mb), which gave results consistent with each other within the error bars.

The observation operator is based on the assumption that observations of the first moment of mass and the total column and the corresponding error covariance matrix are available. The retrievals (after application of the transformation described in Section 3.2) are consistent with this assumption, and the variability which is taken into account in the error estimates should not be included into the observation operator. Evaluating the centre of mass does not depend strongly on the model resolution.

*The sentence on lines 134-135 is extremely problematic as it seems to show a fundamental misunderstanding about assimilation theory: "Finally, the vector y of observations is given by the possibly*

*non-linear observation operator H as y=H(x)+ε where ε denotes the observation error." H(x) is the model state in observation space. y-H(x) is the observation departure. Here ε does not denote only the observation error but all possible errors, including the model and representatitvity errors. Most importantly, y is not "given" by H, it is simulated by H! This awful confusion continues in equation (3) which does not provide y but the (total component of) H(x).*

The sentence has been rephrased.

*It appears from lines 307-312 that the assimilation algorithm can use both the algebraic solution with explicit computation of HM matrix, and the 4D-Var approach. This should be clearly explained in section 3.1. I assume that Vira and Sofiev (2012) provided all necessary details about this 4D-Var implementation and its adjoint model (e.g., was it generated automatically or manually? Is it as detailed as the forward model? How was it verified?). If that is the case, an additional reference and a few words may suffice. But if that was not the case, or if the 4D-Var implementation changed a lot (beyond the developments described in sections 3.2 and 3.3), then this should be fully described (since appropriate for the GMD journal).*

The adjoint model is same as in Vira and Sofiev, (2012). The difference is that in the 2012 study the emission adjustment was multiplicative, horizontally variable but vertically constant, while in the current study, the adjustment is additive and vertically variable but confined to a single column. This corresponds to a different summation of the adjoint variable when evaluating the gradient, but the adjoint code does not need to be changed. A remark has been added to the revised manuscript.

$$

where $\mathbf{f}$ denotes the emission, $\mathbf{x} = \mathbf{M}\,\mathbf{f}$ and $w_k$ is the thickness of the $k$th model layer. To evaluate the L-curve for Tikhonov-regularisation, the parameter $\alpha^2$ was incremented in discrete steps given by $\alpha_i^2 = 10^7 \cdot 2^{-i}$ for $i = 0,1,2,\dots$ . The L-BFGS-B minimization method with non-negativity constraint was used for both Tikhonov regularisation and the truncated iteration; in the case of Tikhonov regularisation, the iteration was continued until convergence for each $\alpha_i^2$ either until convergence or for maximum of 50 iterations. A zero-valued solution was always used as the first guess in the iteration. With the truncated iteration, the weights $w_k$, required by Eqs. (10) and (11) (9) and (10), are not explicitly included in the cost function. Instead, the same effect is achieved by transforming the parameter vector as $f'_{k,n} = w_k^{1/2} f_{k,n}$.

The point where the L-curve flattens, which is taken as the final solution, was determined numerically. First, the points $\left( \left\| \mathbf{f} \right\|, \left\| \mathbf{H}\,\mathbf{x} - \mathbf{y} \right\| \right)$ are sorted according to increasing $\left\| \mathbf{f} \right\|$. 
[revised manuscript text omitted]

---

## Referee Report (RR1)

My concerns about the assimilation/inversion method have been addressed appropriately by the authors. My misunderstanding was entirely due to the assumption that the assimilation window was 24, 12 or 6 hours which are common choices in Numerical Weather Prediction and Air Quality forecasting systems. Here there is one assimilation window covering the whole duration of the case study (21 days) hence it makes perfect sense to ignore any background term in the cost function.

The confusing paragraphs in Sections 3 and 4 have been corrected, and the revised manuscript is much clearer. In order to address the concerns of the first reviewer, the model results are now also evaluated against CALIOP data. This greatly improves the interest of the paper.

I would only suggest one minor revision in the revised abstract which now states (lines 25-29)

> *For Eyjafjallajökull, the comparison between results with and without assimilation of plume height retrievals shows that the estimated injection height was mostly constrained by the inversion even using only total column retrievals. However, comparison with the profile observations from the CALIOP instrument showed that assimilating the plume height retrievals improved the vertical distribution during episodes when the estimated injection height was not otherwise not sufficiently constrained.*

I think that this is explained in a less optimistic manner, but more clearly, in the revised conclusions (lines 601-603):

> *The comparisons show that assimilating the plume height retrievals reduced the overestimation of injection height during individual periods of 1-3 days. However, for most of the simulated 21 days, the injection height was constrained by meteorological conditions and assimilation of the plume height retrievals had only small impact.*